# Prioritize the Process, Not Just the Outcome: Rewarding Latent Thought Trajectories Improves Reasoning in Looped Language Models

Jonathan Williams [1]   Olga Russakovsky [1]   Esin Tureci [1]

## Abstract

Looped Language Models (LoopLMs) perform multi-step latent reasoning prior to token generation and outperform conventional LLMs on reasoning benchmarks at smaller parameter budgets. However, attempts to further improve LoopLM reasoning with reinforcement learning have failed—standard objectives such as Group Relative Policy Optimization (GRPO) only assign credit to the final latent state, creating a fundamental mismatch with the model's internal computation. To resolve this, we introduce **RLTT (Reward Latent Thought Trajectories)**, a reinforcement learning framework which distributes reward across the full latent reasoning trajectory. RLTT provides dense, trajectory-level credit assignment without relying on external verifiers and can directly replace GRPO with negligible overhead. Across extensive experiments with Ouro-1.4B/2.6B-Thinking under identical training and inference conditions, **RLTT yields statistically significant improvements over GRPO on challenging mathematical reasoning benchmarks, improving mean accuracy over MATH-500, AIME24/26, and BeyondAIME by +5.8% on the 1.4B scale, and +10.9% on the 2.6B scale**. Despite being trained exclusively on mathematics, RLTT also transfers effectively to non-mathematical reasoning benchmarks, demonstrating the effectiveness of trajectory-level credit assignment for reinforcement learning in LoopLMs. Code is available at https://github.com/jonwill8/RLTT.git.

---

[1]Department of Computer Science, Princeton University, Princeton NJ, U.S.A. Correspondence to: Jonathan Williams <jw4199@princeton.edu>.

*Proceedings of the $43^{rd}$ International Conference on Machine Learning*, Seoul, South Korea. PMLR 306, 2026. Copyright 2026 by the author(s).

## 1. Introduction

An emerging trend in Large Language Models (LLMs) emphasizes *latent reasoning*: multi-step internal computations performed before emitting each token (Zhu et al., 2025a). Unlike conventional architectures which execute a single forward pass per output token, latent reasoning architectures allow LLMs to iteratively refine internal representations prior to token generation. This enables additional test-time computation without increasing output length, offering a promising alternative to chain-of-thought reasoning.

Looped Language Models (LoopLMs) provide a concrete architectural realization of latent reasoning (Saunshi et al., 2025). LoopLMs recursively apply shared-weight transformer blocks multiple times before emitting each output token. Notably, the Ouro LoopLM (Zhu et al., 2025b) demonstrates that scaling latent computation in this manner yields strong reasoning performance without incurring the chain-of-thought verbosity commonly observed in explicit reasoning approaches.

Prior attempts to apply reinforcement learning with verifiable rewards (RLVR) post-training to LoopLMs have not succeeded (Zhu et al., 2025b). We posit that these attempts failed because standard policy-gradient objectives such as GRPO only reward the last latent state before token emission, implicitly assuming a single-step decision process. This creates a natural tension with LoopLM computation which unfolds over multiple internal refinement steps.

In this work, we introduce RLTT (Reward Latent Thought Trajectories) - a novel reinforcement learning framework built on a simple insight: if a LoopLM reasons across multiple latent iterations, the learning signal should reward that entire trajectory, not just its endpoint. RLTT achieves this by aggregating the LoopLM's next-token distributions across all internal loops and using this aggregate to shape the policy gradient. This yields denser credit assignment, aligns reinforcement learning with the model's iterative computation, and eliminates the need for external verifiers. **In summary:**

- We introduce RLTT, a novel reinforcement learning framework which aligns RLVR with the multi-step latent computation of looped language models.

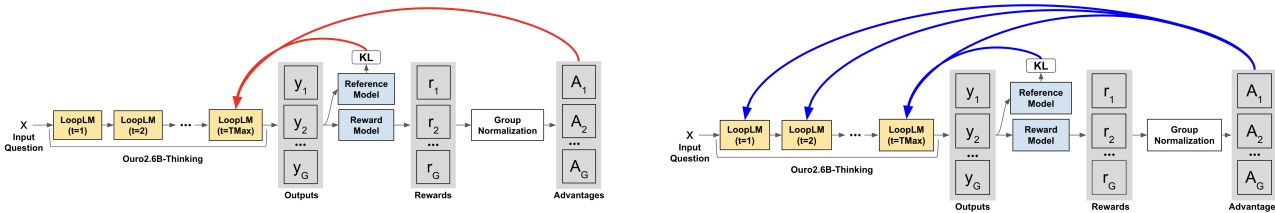

*Figure 1.* Comparison of GRPO vs. RLTT credit assignment. **Left:** GRPO only forms a direct relation between reward and the final loop's predicted next token distribution, creating a credit assignment bottleneck. **Right:** RLTT resolves this issue by rewarding the entire latent thought trajectory (i.e., forming a direct relation between reward and *every* loop's predicted next token distribution).

- We evaluate RLTT on mathematical (MATH-500, AIME24/26, BeyondAIME, GSM8k) and non-mathematical (ARC-C, MMLU-ST, GPQA, MBPP) reasoning benchmarks, realizing consistent performance gains over GRPO.

- We identify that RLTT's core advantage over GRPO is superior token efficiency, and provide a theoretical basis explaining *why* RLTT reaches correct answers in fewer tokens.

## 2. Related Work

**Latent Reasoning Language Models.** A growing set of architectures enable LLMs to perform latent reasoning through multiple internal refinement steps before emitting a token. Looped transformer models such as Ouro (Zhu et al., 2025b) show that repeatedly reusing the same transformer blocks induces emergent "latent thoughts" and test-time reasoning gains comparable to chain-of-thought reasoning. Similarly, deep recurrent-depth transformers such as Huginn (Geiping et al., 2025) repeatedly apply a shared core block, yielding substantial improvements on reasoning tasks by scaling test-time computation in latent space. These models highlight the effectiveness of latent iterative computation as an alternative to explicit token-level reasoning.

**Continuous and Compressed Chain-of-Thought.** Beyond recurrence, several other methods explore reasoning in continuous latent spaces. Coconut (Hao et al., 2025) replaces discrete chain-of-thought tokens with differentiable continuous thoughts, enabling branching-style reasoning and improved planning behavior. CODI (Shen et al., 2025) distills explicit CoT traces into a compact continuous form, while Compressed CoT (CCoT) (Cheng & Durme, 2024) generates a short sequence of contentful latent tokens that encode longer reasoning chains. These works collectively demonstrate that latent-space reasoning can match or exceed explicit CoT.

**Reinforcement Learning for Latent Reasoning LLMs.** Reinforcement Learning (RL) methods such as GRPO have not lead to substantial reasoning gains for Latent Reasoning LLMs. The authors of Ouro - the current state-of-the art

LoopLM - noted that RL was unable to induce significant performance gains over SFT (Zhu et al., 2025b). LSRL introduces per-depth process rewards for the recurrent-depth LLM Huginn by decoding each intermediate latent state and scoring it with GPT-4.1 nano, yielding improvements of +4.27% on GSM8K and +2.06% on MathQA (Ren, 2025). However, decoding intermediate latent states into text and API calling an external grader model introduces significant computational and implementation overhead.

These limitations motivate an alternative approach that assigns credit directly within latent computation and without external supervision. RLTT is purposefully developed for LoopLMs because their architecture yields a next-token distribution at every loop, enabling trajectory-level credit assignment without intermediate decoding. We experimentally evaluate RLTT on Ouro model, to the best of our knowledge, the only open source LoopLM at the time of writing.

## 3. RLTT: Reward Latent Thought Trajectories

### 3.1. Definitions

Let $x$ be the context (prompt) and $y = [y_1, \ldots, y_L]$ be the LoopLM's generated response sequence of length $L$. Suppose that the LoopLM loops for $T_{\max}$ iterations before outputting the next token. Let $h_j^{(t)}$ be the hidden state for the $j$th token after loop iteration $t$. Let $g$ be the language modeling head of the LoopLM which projects token hidden states to the vocabulary dimension. The overall auto-regressive policy induced by the LoopLM (parametrized by $\theta$) is:

$$\pi_\theta(y \mid x) = \prod_{j=1}^{L} P_\theta^{(T_{\max})}(y_j \mid x, y_{<j}), \quad \text{where:}$$

$$P_\theta^{(T_{\max})}(y_j \mid x, y_{<j}) = \text{Softmax}\big(g(h_j^{(T_{\max})})\big) \quad (1)$$

i.e., sampling uses only the terminal-loop "final thought distribution" $P_\theta^{(T_{\max})}$, while intermediate "latent thought distributions":

$$P_\theta^{(t)}(y_j \mid x, y_{<j}) = \text{Softmax}\big(g(h_j^{(t)})\big),$$
$$t = 1, \ldots, T_{\max} - 1 \quad (2)$$

are normally treated as unobserved computation.

## 3.2. RLTT Policy Gradient

Let $r$ be an outcome-based reward function of the ground truth answer $a$ (i.e., it outputs 1 if $y$ matches $a$, 0 otherwise). Consider sampling $g$ rollouts $y_1, y_2, \cdots y_g$ for a given prompt $x$, and computing associated rewards $r_1, r_2, \cdots r_g$. A standard REINFORCE-style policy gradient takes the form:

$$\nabla_\theta J_{\text{standard}}(\theta) = \mathbb{E}_{x \sim \mathcal{D}, \{y_i\}_{i=1}^g \sim \pi_\theta(\cdot | x)}$$

$$\left[ \frac{1}{g} \sum_{i=1}^g \frac{1}{|y_i|} \sum_{j=1}^{|y_i|} \nabla_\theta \log P_\theta^{(T_{\max})}(y_{i,j} \mid x, y_{i,<j}) \hat{A}_i \right] \quad (3)$$

where $\hat{A}_i$ is the advantage computed for the $y_i$ rollout via $\hat{A}_i = \frac{r_i - \text{mean}(\{r_1, r_2, \cdots r_g\})}{\text{std}(\{r_1, r_2, \cdots r_g\})}$. Because this standard REINFORCE-style policy gradient is only a direct function of $P_\theta^{(T_{\max})}$, it treats each token as if it were generated in a single decision step, even though a LoopLM actually computes a full latent thought trajectory $h_j^{(1)} \to \cdots \to h_j^{(T_{\max})}$ before emitting $y_j$.

RLTT instead distributes credit across the entire latent thought trajectory by replacing the single log-probability $\log P_\theta^{(T_{\max})}$ with a weighted sum over loops:

$$\nabla_\theta J_{\text{RLTT PG}}(\theta) = \mathbb{E}_{x \sim \mathcal{D}, \{y_i\}_{i=1}^g \sim \pi_\theta(\cdot | x)}$$

$$\left[ \frac{1}{g} \sum_{i=1}^g \frac{1}{|y_i|} \sum_{j=1}^{|y_i|} \sum_{t=1}^{T_{\max}} \omega_t \nabla_\theta \log P_\theta^{(t)}(y_{i,j} \mid x, y_{i,<j}) \hat{A}_i \right]$$
$$(4)$$

where $\omega_t \geq 0$ are loop weights satisfying $\sum_{t=1}^{T_{\max}} \omega_t = 1$.

From a credit-assignment viewpoint, RLTT: (i) enforces the gradient is a direct function of the latent thought distribution from all loops, preventing the reward signal from having to back-propagate solely through the final thought distribution of the terminal loop, and (ii) enforces that the entire latent thought distribution trajectory $P_{\theta_j}^{(1)} \to \cdots \to P_{\theta_j}^{(T_{\max})}$ is aligned with the rewarded outcome - encouraging the LoopLM for quick convergence to high advantage thinking patterns.

**Adding Reference Policy KL regularization.** To preserve the general language abilities of $\pi_\theta$, we include a KL-divergence regularization term in the final RLTT objective:

$$J_{\text{RLTT}}(\theta) = J_{\text{RLTT PG}}(\theta) + \beta D_{\text{KL}}(\pi_\theta \| \pi_{\text{ref}}) \quad (5)$$

where:

$$J_{\text{RLTT PG}}(\theta) = -\mathbb{E}_{x \sim \mathcal{D}, \{y_i\}_{i=1}^g \sim \pi_\theta(\cdot | x)}$$

$$\left[ \frac{1}{g} \sum_{i=1}^g \frac{1}{|y_i|} \sum_{j=1}^{|y_i|} \sum_{t=1}^{T_{\max}} \omega_t \log P_\theta^{(t)}(y_{i,j} \mid x, y_{i,<j}) \hat{A}_i \right] \quad (6)$$

$$D_{\text{KL}}(\pi_\theta \| \pi_{\text{ref}}) = \mathbb{E}_{x \sim \mathcal{D}, \{y_i\}_{i=1}^g \sim \pi_\theta(\cdot | x)}$$

$$\left[ \frac{1}{g} \sum_{i=1}^g \frac{1}{|y_i|} \sum_{j=1}^{|y_i|} \right.$$

$$\left. D_{\text{KL}}\left( P_\theta^{(T_{\max})}(\cdot \mid x, y_{i,<j}) \| P_{\text{ref}}^{(T_{\max})}(\cdot \mid x, y_{i,<j}) \right) \right] \quad (7)$$

The reference model $\pi_{\text{ref}}$ is a frozen copy of $\pi_\theta$ from before any RLVR post-training.

## 4. Method

RLTT serves as a direct GRPO alternative for LoopLMs. It only assumes that the latent thought trajectory $h_j^{(1)} \to h_j^{(2)} \to \cdots \to h_j^{(T_{\max})}$ of an arbitrary $j$th token can be extracted and mapped to a latent thought distribution trajectory as in (2).

### 4.1. Loop Weighting Strategies

RLTT requires only a choice of loop weights $\{\omega_t\}_{t=1}^{T_{\max}}$, which determine how strongly each latent thought distribution is credited. Weights in all cases are deterministic given the total loop count.

**Exit PDF:**
$$\omega_t = p_{\text{exit}}(t \mid x).$$

This weighting is applicable for LoopLMs such as Ouro which utilize early-exit mechanisms to efficiently terminate looping. Each latent thought is credited the probability produced by Ouro's learned exit head that computation terminates at the $t^{\text{th}}$ loop.

**Progressive:**
$$\omega_t = \frac{t^\alpha}{\sum_{s=1}^T s^\alpha}, \qquad \alpha \geq 0. \quad (8)$$

This weighting ensures that later loops receive more weight, reflecting the intuition that later refinements are closer to the true next token distribution.

**Uniform:**
$$\omega_t = \frac{1}{T_{\max}}, \qquad t = 1, \ldots, T_{\max}. \quad (9)$$

**Algorithm 1** RLTT: Reward Latent Thought Trajectories

**Input:** Prompts $\{x_i\}_{i=1}^{N}$, policy parameters $\theta$, reference parameters $\theta_{\text{ref}}$, rollouts per prompt $g$, max loop depth $T_{\max}$, loop weights $\{\omega_t\}_{t=1}^{T_{\max}}$, KL coefficient $\beta$
**Output:** Updated parameters $\theta_{\text{new}}$
1: **for** $i = 1$ **to** $N$ **do**
2:     Sample $g$ rollouts $\{y_{i,k}\}_{k=1}^{g} \sim \pi_\theta(\cdot \mid x_i)$ (Eq. 1)
3:     **for** $k = 1$ **to** $g$ **do**
4:         Compute reward $r_{i,k} \leftarrow R(a_i, y_{i,k})$
5:     **end for**
6:     Compute advantages $\{\hat{A}_{i,k}\}_{k=1}^{g}$ from $\{r_{i,k}\}_{k=1}^{g}$
7:     **for** $k = 1$ **to** $g$ **do**
8:         **for** $j = 1$ **to** $|y_{i,k}|$ **do**
9:             **for** $t = 1$ **to** $T_{\max}$ **do**
10:                 Record per-loop log-probability $\ell_{i,k,j,t} = \log P_\theta^{(t)}(y_{i,k,j} \mid x_i, y_{i,k,<j})$
11:             **end for**
12:             Record $P_\theta^{(T_{\max})}(\cdot \mid x_i, y_{i,k,<j})$ KL divergence to reference policy $P_{\theta_{\text{ref}}}^{(T_{\max})}(\cdot \mid x_i, y_{i,k,<j})$ (Eq. 7)
13:         **end for**
14:     **end for**
15: **end for**
16: Compute the RLTT objective $J_{\text{RLTT}}(\theta)$ (Eq. 5)
17: $\theta_{\text{new}} \leftarrow \text{OPTIMIZERSTEP}(\theta, \nabla_\theta J_{\text{RLTT}}(\theta))$

In this weighting schema every loop is treated as an equally valid draft model. This encourages the model to form the correct next token distribution as early as possible and maintain it across the latent thought trajectory.

### 4.2. RLTT Algorithm

**Computational cost.** RLTT introduces negligible additional compute relative to GRPO: the per-loop logits are already produced during the LoopLM forward pass, and RLTT's only extra arithmetic is a weighted sum across loops (6), which is linear in $T_{\max}$. The practical overhead is instead memory, which scales linearly with the number of loops because RLTT must retain per-loop log-probabilities to form the trajectory-weighted objective. Under fixed GPU memory, this increased footprint reduces the maximum number of tokens that can be packed per GPU per optimization step (i.e., `ppo_max_token_len_per_gpu`). In our runs, we were forced to set `ppo_max_token_len_per_gpu` to 8192 (half of GRPO's value) and compensate via additional mini-steps.

## 5. Experiments

**Experimental Context.** We evaluate whether RLTT improves mathematical reasoning performance in LoopLMs.

We use Ouro as the base architecture, and compare RLTT against GRPO under strictly compute-matched training conditions: both methods are trained on the same MATH (Hendrycks et al., 2021b) samples, with identical rollout budgets, optimization settings, reward functions, and advantage normalization. We use exit-probability weighting for RLTT, leveraging Ouro's learned halting signal as a proxy for internal confidence - loops with lower exit probability are treated as less reliable and receive proportionally less credit. Additional experimental details are provided in A.5.

### 5.1. Training Dynamics

#### 5.1.1. REWARD EVOLUTION

Figure 2 shows the evolution of reward during training. We observe that RLTT achieves consistently higher reward than GRPO, with the gap emerging early and widening steadily throughout training.

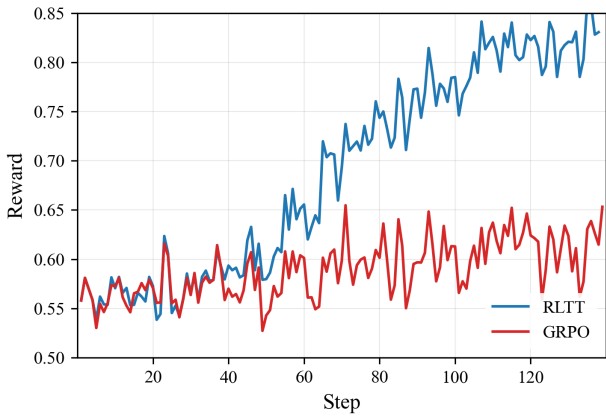

*Figure 2.* **Reward evolution during MATH training.** Mean reward (binary correctness) over training steps under RLTT & GRPO. RLTT achieves consistently higher reward than GRPO throughout training, with the performance gap emerging within the first 40 steps and widening steadily thereafter.

This improvement stems directly from trajectory-level credit assignment. GRPO attributes reward only to the final latent state before token emission, forcing the learning signal to back-propagate through multiple latent refinements. RLTT instead distributes credit across the entire reasoning trajectory, allowing intermediate steps to receive meaningful gradients. This reduces the effective credit-assignment horizon, producing more informative updates and accelerating policy improvement. Because both methods are trained under identical settings, the observed gains indicate that RLTT offers a genuine improvement in learning.

#### 5.1.2. RESPONSE LENGTH EVOLUTION

Figure 3, examines the evolution of response length during training, providing insight into how RLTT alters generation

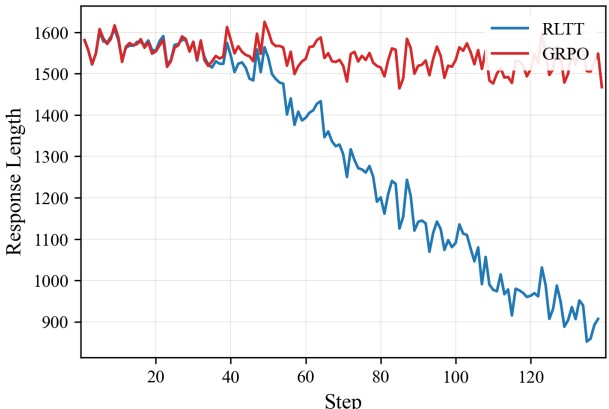

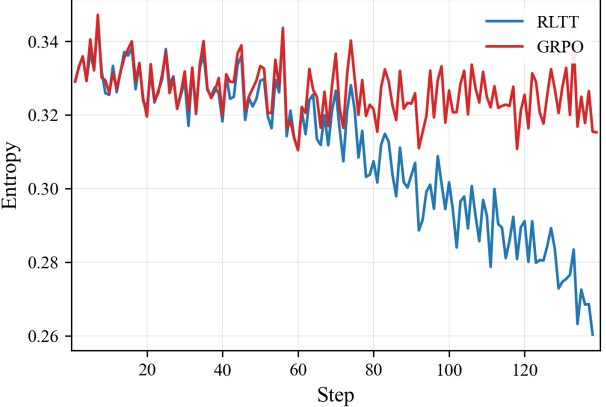

*Figure 3.* **Response length evolution during MATH training.** Mean generated response length over training steps. RLTT-trained policies converge to substantially shorter responses than GRPO, despite the reward function depending exclusively on final-answer correctness with no explicit brevity incentive.

*Figure 4.* **Output token entropy evolution during MATH training.** Mean entropy of the terminal-loop next-token distribution over training steps. RLTT exhibits a steeper and more sustained entropy reduction than GRPO, reflecting increased model confidence as correct reasoning trajectories stabilize.

behavior. A key observation is that RLTT steadily converges to shorter responses over training. Crucially, this effect cannot be attributed to any explicit incentive for brevity: the reward function depends exclusively on final answer correctness, and early termination or concise formatting alone provides no advantage. Instead, reduced response length emerges as a downstream consequence of optimizing the RLTT training objective.

Under RLTT, intermediate latent states are directly aligned with the final outcome, encouraging the model for faster internal convergence to correct reasoning patterns. The observed response length reduction indicates that RLTT induces more efficient latent reasoning - thereby helping eliminate unproductive overthinking and redundant verification, a phenomenon commonly observed in explicit reasoning models (Wei et al., 2026).

| Method | Min / Step | Total Time (hrs) | Rel. Cost |
|--------|------------|------------------|-----------|
| GRPO | 23.3 ± 8.31 | 54.42 | 1.00× |
| RLTT | 21.1 ± 9.87 | 49.05 | 0.90× |

*Table 1.* **MATH training time comparison.** Both GRPO & RLTT are trained for 140 total optimization steps on identical hardware. Min / Step reports the time mean and standard deviation across all training steps. RLTT achieves a 10.0% reduction in training time.

RLTT's emergent response length shortening translates into tangible time savings. As shown in Table 1, RLTT induces a nontrivial 10% reduction in training time. However, this efficiency gain should be interpreted as a secondary effect of improved latent reasoning, not as an optimization target in itself.

### 5.1.3. ENTROPY EVOLUTION

Figure 4 tracks the evolution of output token entropy during training, shedding light on how RLTT affects the model's uncertainty. The large entropy differential, paired with the observed reward improvement and shorter responses raises a natural question - does RLTT suffer from entropy collapse?

We argue that precisely because entropy smoothly decreases in tandem with the reward increase/response length decrease, the observed entropy reduction should be interpreted as controlled confidence rather than degeneration: as correct reasoning trajectories stabilize, fewer alternatives remain plausible. To concretely rule out entropy collapse, we provide a dedicated Pass@k analysis in Appendix A.4, which shows that RLTT retains strong diversity and favorable scaling under increased sampling.

### 5.2. Math Benchmarks

**Evaluation Protocol:** To investigate if RLTT's training improvements generalize to both in-distribution and out-of-distribution mathematical reasoning tasks, we perform zero-shot evaluation on MATH-500 (Hendrycks et al., 2021b), AIME24 (Zhang & Math-AI, 2024), AIME26 (Dekoninck et al., 2026), BeyondAIME (ByteDance, 2025), and GSM8K (Cobbe et al., 2021). All evaluations use deterministic decoding, exact-match answer parsing, and fixed benchmark-specific inference budgets.

**Inference Budgets.** Because GRPO-trained policies produce longer math solutions than RLTT (as evidenced by 5.1.2), we evaluate GRPO with an expanded decode budget on the math benchmarks. Specifically, GRPO is given a 3072-token budget for all math evaluations. RLTT is evaluated with 2048 tokens for MATH-500, 512 tokens for

| Model | Math Benchmarks | | | | | | Non-Math Benchmarks | | | | |
|---|---|---|---|---|---|---|---|---|---|---|---|
| | MATH-500 | AIME24 | AIME26 | BeyondAIME | GSM8K | Avg. | ARC-C | MMLU-ST | GPQA | MBPP | Avg. |
| DeepSeekR1-1.5B | 60.6 | 3.33 | 3.33 | 2.0 | 77.0 | 29.3 | 55.5 | 54.5 | 6.57 | 30.4 | 36.7 |
| DeepSeekR1-7B | 70.4 | 16.7 | 6.67 | 3.0 | 89.0 | 37.2 | 82.5 | 74.4 | 11.1 | 49.7 | 54.4 |
| Qwen3-1.7B | 54.2 | 0.0 | 0.0 | 0.0 | 80.1 | 26.9 | 84.0 | 71.4 | 4.55 | 43.7 | 50.9 |
| + GRPO | 54.6 | 0.0 | 0.0 | 0.0 | 81.3 | 27.2 | 83.4 | 71.2 | 5.1 | 44.7 | 51.1 |
| Qwen3-4B | 63.0 | 3.33 | 3.33 | 0.0 | 88.0 | 31.5 | 90.5 | 77.1 | 11.1 | 56.5 | 58.8 |
| + GRPO | 62.2 | 3.33 | 3.33 | 0.0 | 89.8 | 31.7 | 91.5 | 77.6 | 9.1 | 56.5 | 58.7 |
| Ouro1.4B-Thinking | 73.2 | 16.7 | 13.3 | 4.0 | 90.7 | 39.6 | 90.7 | 78.4 | 12.1 | 53.6 | 58.7 |
| + GRPO | 77.4 | 16.7 | 16.7 | 6.0 | 91.7 | 41.7 | 90.7 | 81.1 | 14.7 | 51.6 | 59.5 |
| + RLTT | 81.2 | 26.7 | 20.0 | 12.0 | 90.3 | 46.0 | 91.9 | 83.8 | 26.8 | 56.7 | 64.8 |
| Ouro2.6B-Thinking | 75.6 | 13.3 | 6.67 | 5.0 | 93.6 | 38.8 | 93.6 | 84.4 | 18.7 | 61.3 | 64.5 |
| + GRPO | 79.0 | 16.7 | 16.7 | 6.0 | 93.9 | 42.5 | 93.7 | 86.1 | 19.7 | 61.3 | 65.2 |
| + RLTT | **86.0** | **33.3** | **26.7** | **16.0** | **94.0** | **51.2** | **94.4** | **89.6** | **38.4** | **64.6** | **71.8** |

*Table 2.* **Performance comparison on mathematical and non-mathematical reasoning benchmarks.** Best results are **bolded**, second-best are underlined. RLTT outperforms GRPO across all benchmarks. Statistical significance analysis A.2 confirms that RLTT's improvements over GRPO are significant ($p < 0.05$) across 7/9 benchmarks at the 1.4B scale, and 8/9 benchmarks at the 2.6B scale.

GSM8K, and 3072 tokens for the remaining benchmarks.

**Evaluation Analysis.** Table 2 summarizes performance across all math benchmarks. RLTT consistently and substantially outperforms GRPO, achieving gains of **+7.0%** on MATH-500, **+16.6%** on AIME24, **+10.0%** on AIME26, and **+10.0%** on BeyondAIME on the 2.6B scale, and **+3.8%** on MATH-500, **+10.0%** on AIME24, **+3.7%** on AIME26, and **+6.0%** on BeyondAIME on the 1.4B scale. Ouro-1.4B+RLTT reaches 46.0% mean benchmark accuracy, outperforming both Qwen3-1.7B+GRPO (27.2%) and Qwen3-4B+GRPO (31.7%), while Ouro-2.6B+RLTT further improves to 51.2%, demonstrating that RLTT compares favorably to non-looped GRPO baselines on mathemeatical reasoning tasks at the 3072 token budget scale.

The gains on AIME24, AIME26, and BeyondAIME are particularly informative. These benchmarks represent the most challenging reasoning regimes in our evaluation suite, requiring sustained multi-step reasoning under strict inference constraints. In these settings, GRPO frequently fails to reach a valid solution before exhausting the token budget. RLTT's improvements therefore primarily arise from enabling the model to arrive at correct solutions without superfluous token-level exploration. This interpretation is consistent with the training dynamics analyses in Section 5.1, which show that RLTT achieves correct latent reasoning trajectories in fewer decoding steps.

Importantly, *RLTT's advantage is not limited to the specific inference budget used during training.* Appendix A.1 evaluates robustness across a wide range of decode lengths on MATH-500, spanning heavily constrained (1024 tokens) to substantially looser (4096 tokens) regimes. RLTT consistently outperforms GRPO across this entire spectrum, including at 4096 tokens—far beyond the 2048-token train-ing horizon. This indicates that RLTT does not overfit to a particular decode length, but instead learns reasoning policies that generalize across inference-time compute regimes.

Further insight comes from analyzing reasoning capacity at the loop level. Appendix A.6 shows that *RLTT outperforms GRPO at every evaluated loop count, with especially large margins under 1–2 loop regimes.* This result directly supports the claim that RLTT improves the effectiveness of early latent reasoning iterations, explaining its robustness under tight token and loop budgets.

To verify that these improvements are not artifacts of deterministic decoding, Appendix A.2 reports the results of a statistical significance analysis using paired t-tests. *RLTT remains statistically significantly superior to GRPO across all math benchmarks at the 2.6B scale, and 3/5 math benchmarks at the 1.4B scale, confirming that the observed gains reflect genuine improvements in reasoning.*

Finally, Appendix A.3 shows that *RLTT's performance is largely insensitive to the precise loop-weighting strategy used to distribute credit*, indicating that the gains stem from exposing the reinforcement learning signal to the full latent trajectory rather than from carefully tuned credit schedules.

> **Takeaways.** RLTT substantially improves mathematical reasoning in LoopLMs. RLTT trains policies that reach correct solutions more robustly under strict inference constraints – an advantage that persists across various decode budgets, loop capacities, evaluation protocols, and weighting schemas.

**From Mathematical to General Reasoning Benchmarks.** We next investigate whether RLTT engenders gains which are specific to symbolic mathematics, or instead reflect a

more general improvement in latent reasoning. To this end, we extend our evaluation to a set of non-mathematical reasoning benchmarks spanning logical inference, factual recall, multi-domain question answering, and coding. Importantly, these tasks are never observed during training: all models were trained exclusively on the MATH training split.

### 5.3. Non-Math Benchmarks

**Evaluation Protocol.** We perform zero-shot evaluation on ARC-C (Clark et al., 2018), MMLU-ST (Hendrycks et al., 2021a), GPQA (Rein et al., 2023) and MBPP (Austin et al., 2021). As in the math benchmarks, all models are evaluated without any additional fine-tuning, using deterministic decoding and exact-match answer parsing.

**Inference Budgets.** To ensure evaluation occurs within the same optimized reasoning regime learned during GR-PO/RLTT training, we match the maximum generation length used and employ a 2048-token decode budget for ARC-C, MMLU-ST, GPQA, and MBPP.

**Evaluation Analysis.** Table 2 summarizes zero-shot performance across all non-mathematical benchmarks. Despite being trained exclusively under mathematical supervision, RLTT consistently outperforms both GRPO, achieving gains of **+0.7%** on ARC-C, **+3.5%** on MMLU-ST, **+18.7%** on GPQA, and **+3.3%** on MBPP on the 2.6B scale, and **+1.2%** on ARC-C, **+2.7%** on MMLU-ST, **+12.1%** on GPQA, and **+5.1%** on MBPP on the 1.4B scale. Ouro-1.4B+RLTT reaches 64.8% mean non-math benchmark accuracy, outperforming both Qwen3-1.7B+GRPO (51.1%) and Qwen3-4B+GRPO (58.7%), while Ouro-2.6B+RLTT further improves to 71.8%, demonstrating that RLTT compares favorably to non-looped GRPO baselines on general reasoning tasks at the 2048 token budget scale.

Importantly, these gains are not artifacts of deterministic decoding. Appendix A.2 reports the results a statistical significance analysis using paired t-tests. *RLTT achieves statistically significant improvements over GRPO on all evaluated non-math benchmarks at both the 2.6B and 1.4B scales. This confirms that the improvements observed in Table 2 reflect robust differences in reasoning behavior rather than sensitivity to a particular decoding strategy.*

The most pronounced gain is observed on GPQA, where RLTT nearly doubles performance relative to GRPO. GPQA requires multi-hop factual reasoning, making it particularly sensitive to how efficiently reasoning trajectories are stabilized. GRPO frequently fails in this regime by expending the majority of the decode budget on unproductive exploration, whereas RLTT more reliably converges to a coherent solution path within the available token horizon. This mirrors the failure mode observed on AIME24/26 and BeyondAIME in the mathematical setting, reinforcing the interpretation

that RLTT's fundamental advantage stems from improved reasoning efficiency.

On ARC-C, MMLU-ST, and MBPP – benchmarks where baseline Ouro performance is already relatively strong – RLTT yields smaller but consistent gains. We emphasize that all improvements emerged despite the absence of any non-math supervision during training, emphasizing that RLTT modifies *how* the model reasons rather than *what* it learns to solve.

> **Takeaways**. RLTT induces a transferable improvement in latent reasoning that generalizes beyond mathematics. RLTT-trained policies reach correct conclusions more efficiently in STEM domains spanning logical reasoning, factual recall, multi-domain question answering, and code generation.

## 6. Why Does RLTT Work?

RLTT's strong performance raises a natural question: why does it work? We argue that RLTT's improvements stem from resolving a fundamental credit-assignment mismatch between terminal-only objectives and the multi-step latent computation intrinsic to LoopLMs. This resolution manifests in two complementary ways: more token-efficient reasoning at inference time, and richer gradient signal during training.

### 6.1. Token-Efficient Reasoning

Under terminal-only objectives such as GRPO, correctness is enforced only at the final latent state, permitting learned policies to rely on extended token-level reasoning and late-stage correction. In contrast, distributing reward across latent reasoning iterations favors policies that consistently internally converge to correct solutions, and thus don't require superfluous token-level reasoning. Formally, we show in theorem A.5 of Appendix A.10 that, under three reasonable assumptions:

1. **Loop Refinement Decreases Uncertainty**: Let $V(\cdot)$ be any nonnegative *uncertainty / variance functional* on categorical distributions (e.g., entropy or variance). Define the per-token loop uncertainty

$$V_\theta^{(t)}(j) := V\left(P_\theta^{(t)}(\cdot \mid x, y_{<j})\right) \geq 0.$$

We asssume for every $j$ and every context $(x, y_{<j})$, uncertainty is non-increasing with loop depth:

$$V_\theta^{(1)}(j) \geq V_\theta^{(2)}(j) \geq \cdots \geq V_\theta^{(T_{\max})}(j).$$

2. **Diminishing Returns of Extra Decoded Tokens**: Let $S : \mathbb{Z}_{\geq 0} \to [0, 1]$ denote the best achievable expected

terminal reward under a maximum length constraint:

$$S(L) := \sup_{\pi_\theta:\ \Pr_{y\sim\pi_\theta(\cdot|x)}[L(y)\leq L]=1} \mathbb{E}_{y\sim\pi_\theta(\cdot|x)}[r(y)]$$

where $r(y) \in [0,1]$ depends only on final-answer correctness. We assume that $S(L)$ is non-decreasing and discrete concave in $L$, i.e., the marginal gains

$$\Delta S(L) := S(L+1) - S(L)$$

are non-increasing in $L$.

3. **Approximately Linear Total Uncertainty Cost**

There exist constants $c_{\text{GRPO}}, c_{\text{RLTT}} > 0$ such that, in expectation, each generated response incurs an uncertainty cost approximately linear in response length:

$$\mathbb{E}\left[\sum_{j=1}^{L(y)} C_\theta^{\text{GRPO}}(j) \;\middle|\; L(y) = L\right] \approx c_{\text{GRPO}}\,L$$

$$\mathbb{E}\left[\sum_{j=1}^{L(y)} C_\theta^{\text{RLTT}}(j) \;\middle|\; L(y) = L\right] \approx c_{\text{RLTT}}\,L$$

where $C_\theta^{\text{GRPO}}(j) := V_\theta^{(T_{\max})}(j)$, and $C_\theta^{\text{RLTT}}(j) := \bar{V}_\theta(j) = \sum_{t=1}^{T_{\max}} \omega_t\, V_\theta^{(t)}(j)$. We further assume $c_{\text{RLTT}} \geq c_{\text{GRPO}}$ as $C_\theta^{\text{RLTT}}(j)$ averages per-token uncertainty across all latent loops while $C_\theta^{\text{GRPO}}(j)$ only uses the final-loop uncertainty.

trajectory-level credit assignment reduces the expected number of token decoding steps required to produce a correct solution relative to terminal-only credit assignment.

---

**Theorem A.5 RLTT selects weakly smaller optimal length.** Consider the abstract reward–cost tradeoff

$$\max_{L\in\mathbb{Z}_{\geq 0}}\ S(L) - \phi\, c\, L,$$

Let:

$$L_{\text{GRPO}}^\star \in \arg\max_{L\geq 0}\big[S(L) - \phi\, c_{\text{GRPO}}L\big]$$

$$L_{\text{RLTT}}^\star \in \arg\max_{L\geq 0}\big[S(L) - \phi\, c_{\text{RLTT}}L\big]$$

If $c_{\text{RLTT}} \geq c_{\text{GRPO}}$, then any optimizer satisfies

$$L_{\text{RLTT}}^\star \leq L_{\text{GRPO}}^\star.$$

*Thus, trajectory-level credit assignment induces weakly shorter optimal decoding lengths than terminal-only credit under the same reward structure.*

---

This theoretical result helps explain RLTT's empirical robustness under constrained inference budgets. When decoding length is limited, policies that depend on late-stage correction are brittle as the token budget may be exhausted before the correct answer is achieved. GRPO-trained policies are thus vulnerable in restrictive inference regimes while RLTT-trained policies - due to their stronger latent reasoning and reduced decode token requirements - remain robust.

### 6.2. Richer Gradient Signal

The same credit-assignment mechanism that improves inference efficiency also enhances learning dynamics. To quantify this, we measured Gradient Signal-to-Noise Ratio (GSNR) across MATH-500, AIME24, BeyondAIME, and GSM8K. RLTT yields statistically significant GSNR improvements on AIME24 and BeyondAIME — the hardest benchmarks where credit assignment is most challenging. On these tasks, terminal-only objectives concentrate the learning signal at the final latent state, limiting the gradient information available to shape earlier computation. By exposing the full latent thought trajectory to the reward signal, RLTT increases the density of gradient information per rollout, yielding a richer supervisory signal. Full details are provided in Appendix A.7.

---

**Takeaways.** RLTT improves reasoning efficiency by resolving a fundamental credit-assignment mismatch in LoopLMs. Distributing reward across latent thought trajectories leads to provably reduced decoding length, improved robustness under constrained inference budgets, and richer learning signals on challenging benchmarks.

---

## 7. Conclusion

In this work, we introduce Reward Latent Thought Trajectories (RLTT), a framework which aligns reinforcement learning with the multi-step latent computation intrinsic to looped language models. By distributing credit across the entire latent reasoning trajectory, rather than bottlenecking reward at the terminal loop, RLTT provides dense, trajectory-level supervision without relying on external verifiers. Across extensive evaluations under strictly matched training and inference conditions, RLTT consistently outperforms GRPO on challenging mathematical benchmarks, while also exhibiting strong zero-shot transfer to non-mathematical reasoning tasks. Our analyses further reveal that RLTT induces earlier stabilization of correct reasoning trajectories, leading to more decisive generation, improved robustness under restrictive loop/token budgets, and richer gradient signals precisely in the most difficult reasoning regimes.

At the same time, RLTT introduces practical trade-offs. Re-

taining per-loop log-probabilities increases memory footprint, limiting per-GPU token packing, and the method remains specialized to looped architectures rather than standard non-looped architectures. In addition, our experiments use a fixed loop depth during training and inference, which sacrifices Ouro's native ability to adaptively choose when to early-exit and allocate fewer (or more) latent iterations per token based on input difficulty. Future work will explore memory-efficient implementations and extensions that integrate adaptive halting to recover per-token compute allocation while preserving trajectory-level credit assignment.

## Impact Statement

This work aims to advance the field of machine learning by improving how reinforcement learning aligns with latent reasoning in looped language models. While the techniques proposed may contribute to more capable and efficient reasoning systems, we do not foresee any immediate negative societal impacts beyond those commonly associated with general-purpose language models.

## Acknowledgments

We would like to thank the Princeton President's Fellowship and the GEM Fellowship for supporting this work.

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

# A. Appendix

**Appendix Roadmap:**

### A.1. Analysis of Decode Token Budget Robustness

| Method | 1024 tokens | 2048 tokens | 3072 tokens | 4096 tokens |
|---|---|---|---|---|
| Ouro2.6B-Thinking | 38.4 | 67.8 | 75.6 | 79.6 |
| + SFT | 34.6 | 58.2 | 71.8 | 77.6 |
| + GRPO | 42.4 | 71.6 | 79.0 | 80.8 |
| + RLTT | **78.4** | **86.0** | **87.4** | **89.8** |

*Table 3.* Decode-budget robustness investigation on MATH-500.

This ablation evaluates whether performance gains from RLTT persist under both tighter and looser inference-time compute budgets. We restrict evaluation to a maximum of 4096 tokens, as this regime already exceeds the training-time rollout horizon by 2x and suffices to test out-of-distribution inference robustness. Across the full range of decode budgets, RLTT consistently outperforms GRPO, demonstrating robustness at both extremes of inference-time constraint. Under a heavily constrained 1024-token budget, RLTT retains high MATH-500 accuracy (78.4%), whereas GRPO and SFT degrade sharply, indicating strong sensitivity to truncation. This regime highlights RLTT's ability to make decisive progress toward correct solutions early in generation. In contrast, GRPO appears to rely more heavily on extended token-level reasoning traces, which become brittle when inference budgets are restricted.

Importantly, RLTT's advantage is not confined to tight-budget settings. Even under a much looser 4096-token budget—far exceeding the 2048-token horizon used during training—RLTT continues to achieve higher accuracy than GRPO. This demonstrates that RLTT's gains do not arise from overfitting to a particular decode length or training-specific budget, but instead reflect more effective reasoning policies that generalize beyond the training distribution. While the absolute performance gap narrows as GRPO is given sufficient length to amortize its reasoning over longer outputs, RLTT maintains a consistent advantage without requiring additional inference compute.

Overall, these results indicate that RLTT learns policies that are not only more accurate, but also more robust across a wide spectrum of inference-time constraints, from highly restrictive deployment settings to permissive regimes well outside the training distribution. We restrict this analysis to MATH-500, as it is the only math benchmark in our suite where multiple decode budgets are simultaneously valid; for AIME24 and BeyondAIME, shorter budgets lead to near-zero accuracy across all methods due to truncation, while GSM8K performance saturates well below 1024 tokens.

## A.2. Statistical Significance Analysis of RLTT Improvements.

In our main experiments, we follow standard practice and report deterministic decoding results for Pass@1 evaluation. However, deterministic decoding does not admit variance-based statistical significance testing.

To assess the robustness and statistical significance of the performance gains achieved by RLTT over GRPO, we conduct additional sampling-based evaluations that introduce stochasticity at inference time. Specifically, we evaluate both methods using a fixed temperature of $T = 0.2$, while keeping all other evaluation conditions identical across methods. For each benchmark problem, we generate multiple independent samples using 10 different random seeds and compute the average accuracy per task. To highlight RLTT's token efficiency, we permit extended token budgets for GRPO of 3072 on MATH-500/GSM8K compared to RLTT's 2048/512.

We then perform paired t-tests between RLTT and GRPO using matched samples from the same set of questions, following prior work (Yue et al., 2025). Table 4 reports the averaged results on both math and non-math benchmarks, with statistically significant improvements ($p < 0.05$) highlighted in bold.

| Model | Method | Math Benchmarks | | | | | Non-Math Benchmarks | | | |
|---|---|---|---|---|---|---|---|---|---|---|
| | | MATH-500 | AIME24 | AIME26 | BeyondAIME | GSM8K | ARC-C | MMLU-ST | GPQA | MBPP |
| Ouro1.4B-Thinking | GRPO | 76.9±0.8 | 16.0±2.6 | 14.3±3.9 | 7.00±1.8 | **92.4±0.3** | 91.0±0.4 | 80.9±0.4 | 13.4±1.3 | 60.9±0.8 |
| | RLTT | **81.5±0.7** | **20.7±5.4** | 17.3±4.7 | **11.3±1.8** | 90.8±0.4 | **91.8±0.5** | **84.4±0.2** | **26.9±1.6** | **64.8±0.8** |
| Ouro2.6B-Thinking | GRPO | 79.1±0.7 | 17.7±2.3 | 14.7±3.6 | 9.30±1.9 | 94.5±0.3 | 93.5±0.4 | 86.0±0.3 | 19.7±2.0 | 68.0±1.3 |
| | RLTT | **85.9±1.0** | **32.3±4.2** | **27.0±4.0** | **16.4±2.4** | 94.4±0.4 | **94.3±0.3** | **89.9±0.3** | **38.2±1.9** | **73.6±0.6** |

*Table 4.* Statistical significance analysis of RLTT performance improvements over GRPO under sampling-based evaluation. Bold indicates statistically significant improvement over GRPO of the same model size ($p < 0.05$, paired $t$-test).

RLTT achieves statistically significant gains over GRPO on 8/9 benchmarks on the 2.6B scale, and 7/9 benchmarks on the 1.4B scale. These findings indicate that the improvements observed in Pass@1 evaluation are not artifacts of deterministic decoding, but persist under stochastic inference. These results provide additional evidence that RLTT delivers robust and reliable performance improvements over GRPO.

## A.3. Analysis of RLTT Weighting Strategy

| Weighting Strategy | Math Benchmarks | | | | Non-Math Benchmarks | | | |
| --- | --- | --- | --- | --- | --- | --- | --- | --- |
| | MATH-500 | AIME24 | BeyondAIME | GSM8K | ARC-C | MMLU-ST | GPQA | MBPP |
| Uniform | 85.0 | **36.7** | **17.0** | **94.8** | 94.2 | **90.4** | **45.5** | 59.1 |
| Progressive ($\alpha = 1$) | **86.8** | 30.0 | 16.0 | **94.8** | 94.2 | **90.4** | 41.9 | **64.6** |
| Exit-probability | 86.0 | 33.3 | 16.0 | 94.0 | **94.4** | 89.6 | 38.4 | **64.6** |

*Table 5.* Sensitivity of RLTT performance to loop-weighting strategy across reasoning benchmarks.

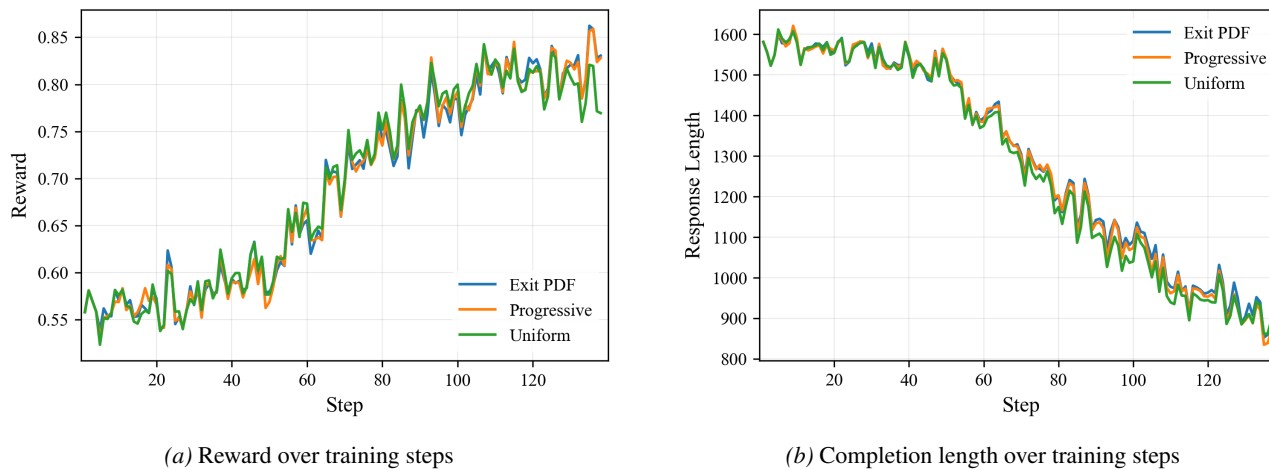

*(a)* Reward over training steps

*(b)* Completion length over training steps

*Figure 5.* Comparison of RLTT weight methods on MATH training dynamics.

Table 5 and Figure 5 analyze the sensitivity of RLTT to the choice of loop-weighting strategy used to distribute credit across latent thought iterations. We compare three natural weighting schemes—uniform, progressive, and exit-probability weighting—across both mathematical and non-mathematical reasoning benchmarks, holding all other training and evaluation conditions fixed.

A key observation is that performance differences across weighting strategies are consistently modest. Across all benchmarks, the variation induced by changing the weighting scheme is small relative to the performance gap between RLTT and GRPO. This indicates that RLTT's gains are not driven by a carefully tuned credit allocation schedule, but instead arise from the more fundamental change of exposing the reinforcement learning signal to the entire latent reasoning trajectory. Once credit is no longer bottlenecked through the terminal loop alone, the precise distribution of that credit across loops becomes a second-order consideration. *This robustness is an important practical property, as it suggests that practitioners need not perform delicate hyperparameter tuning over latent credit allocation in order to benefit from trajectory-level reinforcement learning.*

## A.4. Pass@k Analysis

To assess whether RLTT's lower entropy reflects controlled confidence rather than collapse, we analyze Pass@k performance under stochastic sampling with a fixed temperature of $T = 0.6$ and identical token decode budgets (2048 for MATH-500, 3072 for AIME24/BeyondAIME, and 512 for GSM8K).

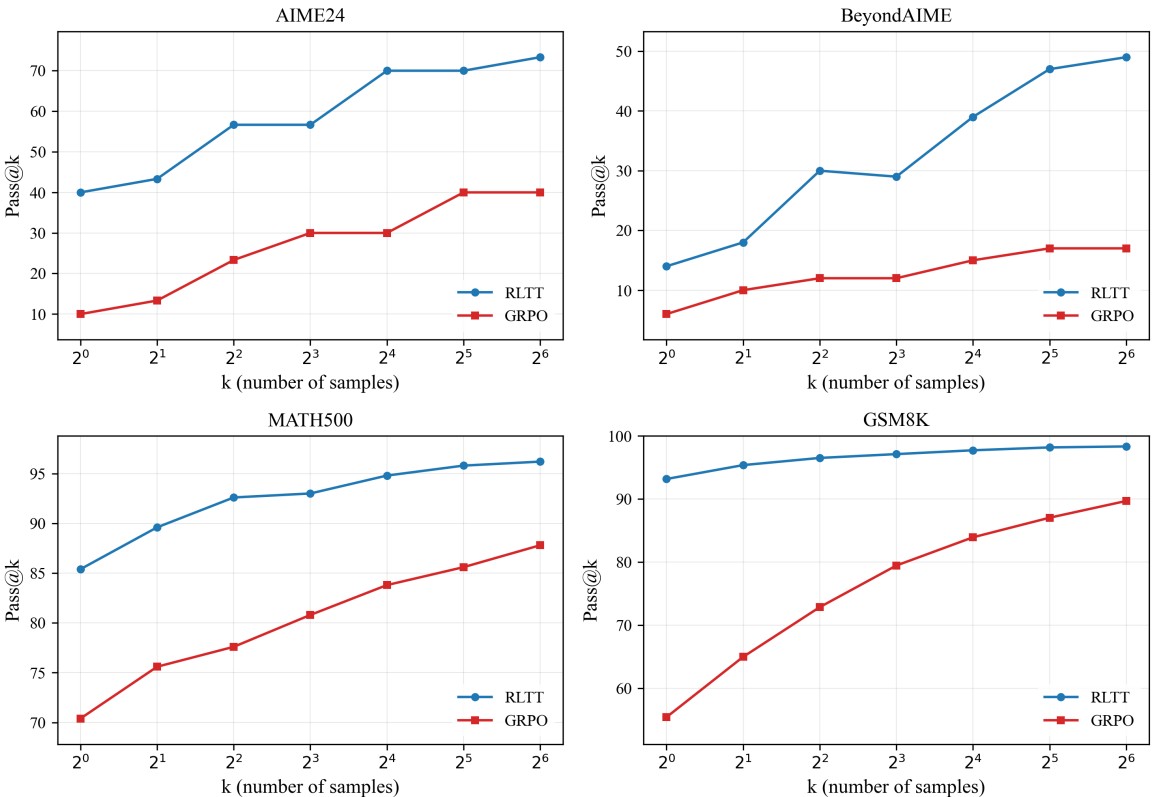

*Figure 6.* Pass@k evaluation under sampling temperature $T = 0.6$. Results compare GRPO and RLTT performance across all benchmarks as the number of sampled solutions $k$ increases.

Figure 6 reports Pass@k accuracy as k increases. Across all benchmarks, RLTT exhibits consistently steeper scaling than GRPO, indicating its gains arise from a richer set of viable reasoning paths rather than a single dominant mode.

This effect is most pronounced on AIME24 and BeyondAIME — the hardest benchmarks where GRPO shows limited improvement with additional samples. RLTT continues accruing substantial gains at higher k, implying a larger fraction of sampled solutions correspond to coherent reasoning processes. On MATH-500, RLTT maintains clear advantages across all k values, while GSM8K saturates rapidly for both methods due to near-ceiling performance.

These trends corroborate the entropy dynamics in 5.1.3: although RLTT exhibits lower output entropy, Pass@k scaling demonstrates this reflects controlled confidence rather than premature collapse. The model becomes more decisive earlier in generation while retaining sufficient stochasticity for effective sampling-based inference. This provides additional evidence that trajectory-level credit assignment concentrates probability mass on correct solution manifolds without eliminating diversity.

## A.5. Further Experimental Details

### A.5.1. TRAINING DETAILS

We use the VERL framework to enable distributed training over 4 H200 140GB GPUs. All Ouro models are trained with full parameter updates and are initialized from the same Ouro-2.6B-Thinking checkpoint. vLLM is used for rollout acceleration.

| Algorithm | GRPO/RLTT |
| --- | --- |
| Loop Iterations | 4 |
| Steps | 140 |
| Prompt Batch Size | 32 |
| # Rollouts per Prompt | 8 |
| Max Generation Length | 2048 |
| Learning Rate | $1 \times 10^{-6}$ |
| LR Scheduler | Constant with Warmup |
| Optimizer | AdamW 8bit |
| Optimizer Momentum | $\beta_1, \beta_2 = 0.9, 0.99$ |
| KL Coefficient | $1 \times 10^{-3}$ |
| Weight Decay | 0.1 |
| Max Gradient Norm | 0.1 |
| Gradient Accumulation Step | 2 |
| Warmup Ratio | 0.1 |
| Precision (WA) | BF16 |
| Reward Formulation | Binary 0-1 |
| Advantage Estimation | Group Normalized |

*Table 6.* Ouro Experimental Hyperparameters.

### A.5.2. MATH BENCHMARK EVALUATIONS

MathVerify is used for answer parsing.

| | |
| --- | --- |
| Prompt / Completion Length for MATH500 | 1024 / 2048 \| 3072 |
| Prompt / Completion Length for AIME24 | 1024 / 3072 |
| Prompt / Completion Length for BeyondAIME | 1024 / 3072 |
| Prompt / Completion Length for GSM8K | 512 / 512 \| 3072 |

*Table 7.* Experiment prompt / completion lengths.

**Ouro Prompt for Math Evaluations**

```
<|im_start|>system
You are a helpful assistant.<|im_end|>
<|im_start|>user

Solve the following math problem. Show your reasoning step by step.
Put your final answer in \boxed{}.
Once you provide the final answer, stop immediately.

Problem: <problem text here><|im_end|>
<|im_start|>assistant
```

### A.5.3. NON-MATH EXPERIMENTS

Pass@1 evaluation with a 5 second timeout is used for MBPP.

| | |
|---|---|
| Prompt / Completion Length for ARC-C | 512 / 2048 |
| Prompt / Completion Length for MMLU-ST | 512 / 2048 |
| Prompt / Completion Length for GPQA | 3072 / 2048 |
| Prompt / Completion Length for MBPP | 4096 / 2048 |

*Table 8.* Experiment prompt / completion lengths.

---

**Ouro Prompt for ARC-C, MMLU-ST, and GPQA Evaluations**

```
<|im_start|>system
You are a helpful assistant that answers multiple choice questions.
Think through the problem step by step,
then give your final answer as a single letter (A, B, C, or D).<|im_end|>
<|im_start|>user
Question: {question}

A. {choice_A}
B. {choice_B}
C. {choice_C}
D. {choice_D}

Think step by step and then provide your answer.
Put your final answer in \boxed{}.
Once you provide the final answer, stop immediately.<|im_end|>
<|im_start|>assistant
```

---

**Ouro Prompt for MBPP Evaluation**

```
<|im_start|>system
You are an expert Python programmer.
Write a Python function that solves the given task.
Make sure your code passes all the provided test cases.<|im_end|>
<|im_start|>user
Task: {prompt}

Test cases that your solution should pass:
{test_str}

Write a Python function that solves this task.
Put your solution in a Python code block.<|im_end|>
<|im_start|>assistant
```

---

### A.5.4. QWEN3 GRPO BASELINES

To address reviewer requests for additional non-looped RL baselines, we train Qwen3+GRPO models at the 1.7B and 4B parameter scales. These baselines test whether standard GRPO post-training on a conventional decoder-only architecture is sufficient to match the improvements obtained by applying trajectory-level credit assignment to LoopLMs.

Both Qwen3 models are trained on the same MATH training data used in the Ouro experiments. We use a learning rate of $1 \times 10^{-4}$ for Qwen3+GRPO, with all other experimental settings matched to the GRPO/RLTT protocol described above, including rollout budget, reward formulation, advantage normalization, optimization length, and evaluation procedure. Because Qwen3 does not expose a loop-wise latent thought trajectory, RLTT is not applicable to these baselines; GRPO therefore serves as the corresponding RLVR comparison.

## A.6. Analysis of Loop-Level Performance

| Dataset | Method | 1 Loop | 2 Loops | 3 Loops | 4 Loops |
|---|---|---|---|---|---|
| MATH-500 | SFT | 27.4 | 61.8 | 63.4 | 58.2 |
|  | GRPO | 32.4 | 66.2 | 70.4 | 71.6 |
|  | RLTT | **37.4** | **81.2** | **84.8** | **86.0** |
| AIME24 | SFT | 0.00 | 6.67 | 13.3 | 13.3 |
|  | GRPO | 0.00 | 10.0 | 16.7 | 16.7 |
|  | RLTT | 0.00 | **23.3** | **26.7** | **33.3** |
| BeyondAIME | SFT | 0.00 | 4.00 | 4.00 | 6.00 |
|  | GRPO | 0.00 | 2.00 | 6.00 | 6.00 |
|  | RLTT | 0.00 | **11.0** | **15.0** | **16.0** |
| GSM8K | SFT | 32.4 | 60.7 | 62.2 | 59.6 |
|  | GRPO | 33.2 | 58.4 | 62.4 | 59.7 |
|  | RLTT | **59.4** | **89.9** | **93.1** | **94.0** |

*Table 9.* Per-loop evaluations on MATH-500, GSM8K, AIME24, and BeyondAIME. RLTT and GRPO are evaluated while varying the maximum number of reasoning loops from 1 to 4. RLTT consistently outperforms GRPO.

**Loop-level Analysis**.

Table 9 compares GRPO and RLTT while varying the number of reasoning loops at evaluation time across MATH-500, GSM8K, AIME24, and BeyondAIME. Across all benchmarks, RLTT consistently outperforms GRPO at every loop count for which non-zero accuracy is achieved.

On MATH-500, RLTT improves 1-loop accuracy by +5.0 points over GRPO and maintains a widening advantage as additional loops are enabled, reaching a +14.4 point improvement at 2 loops and a +14.4 point improvement at 4 loops. These gains indicate that RLTT benefits not only from increased loop capacity but also from more effective utilization of early reasoning iterations.

A similar pattern emerges on GSM8K, where RLTT exhibits particularly strong gains under constrained loop budgets. At 1 loop, RLTT outperforms GRPO by +26.2 points, and this gap further widens to +31.5 points at 2 loops . Even at higher loop counts, RLTT maintains substantial margins, achieving +30.7 and +34.3 point improvements at 3 and 4 loops, respectively. These results suggest that RLTT induces significantly more effective reasoning trajectories when loop capacity is limited.

For the more challenging benchmarks AIME24 and BeyondAIME, both methods achieve zero accuracy at 1 loop, rendering this regime uninformative. However, once additional loops are enabled, RLTT diverges sharply from GRPO. On AIME24, RLTT surpasses GRPO by +13.3 points at 2 loops, +10.0 points at 3 loops, and +16.6 points at 4 loops . On BeyondAIME, RLTT achieves gains of +9.0, +9.0, and +10.0 points at 2, 3, and 4 loops, respectively, reflecting consistent improvements as loop depth increases.

## A.7. Analysis of Gradient Signal to Noise Ratio

Taking inspiration from (Liu et al., 2020) and (Zhu et al., 2025c), we measure the quality of the learning signal produced by the GRPO and RLTT objectives using the gradient signal-to-noise ratio (GSNR). We compute GSNR with respect to the model's final loop latent-thought logits: $z_j^{(T_{\max})}$, where $P_\theta^{(T_{\max})}(y_j \mid x, y_{<j}) = \text{Softmax}(z_j^{(T_{\max})})$.

For a given prompt $p$, we sample $R = 8$ independent responses $\{r_1, \ldots, r_R\}$ using the current policy. Each response is graded to compute advantages, which are then used to form the GRPO or RLTT loss $\mathcal{L}_{p,r}$ for rollout $r$.

For each rollout, we compute the gradient of the loss with respect to all latent-thought logits used in that rollout:

$$g_{p,r} = \nabla_{z_{p,r}} \mathcal{L}_{p,r},$$

where $z_{p,r}$ denotes the $r$th latent thought logit, in the $p$th response generation.

To quantify the consistency of the learning signal across rollouts for a fixed prompt, we compute the mean gradient

$$\mu_p = \frac{1}{R} \sum_{r=1}^{R} g_{p,r},$$

and the corresponding noise

$$\text{noise}_p = \frac{1}{R-1} \sum_{r=1}^{R} \|g_{p,r} - \mu_p\|_2^2.$$

We define the prompt-level gradient signal-to-noise ratio as

$$\text{GSNR}_p = \frac{\|\mu_p\|_2^2}{\text{noise}_p + \epsilon},$$

where $\epsilon = 1.0 \times 10^{-8}$ is a small constant for numerical stability. This procedure is repeated over all $N$ prompts from each evaluation benchmark, producing prompt-level GSNR values $\{\text{GSNR}_p\}_{p=1}^{N}$. We report the overall GSNR as the mean of the log-transformed prompt-level scores:

$$\text{GSNR} = \frac{1}{N} \sum_{p=1}^{N} \log\left(\text{GSNR}_p + \epsilon\right),$$

which provides a stable aggregate measure of gradient signal quality. Table 10 reports GSNR for both GRPO and RLTT over all math evaluation benchmarks. A noticeable pattern emerges: RLTT yields statistically significant GSNR improvements on the hardest benchmarks - AIME24 and BeyondAIME - while differences on MATH-500 are modest and GSM8K exhibits a statistically significant decrease.

| Dataset | Method | GSNR |
|---------|--------|------|
| MATH-500 | GRPO | -14.7 |
| | RLTT | -15.3 |
| AIME24 | GRPO | -15.1 |
| | RLTT | **-11.8** |
| BeyondAIME | GRPO | -16.8 |
| | RLTT | **-13.8** |
| GSM8K | GRPO | **-12.0** |
| | RLTT | -17.0 |

*Table 10.* Gradient Signal-to-Noise Ratio (GSNR) at the latent-thought level. Statistically significant improvements ($p < 0.05$, paired t-test) are shown in **bold**.

These GSNR trends are consistent with how RLTT reshapes the optimization signal. On the most challenging benchmarks AIME24 and BeyondAIME, correct solutions are rare and rewards are correspondingly sparse. Under terminal-only credit assignment, the learning signal is concentrated at the final latent state, limiting the amount of gradient signal available to shape earlier internal computation. By distributing credit across the latent thought trajectory, RLTT exposes a larger fraction of the model's internal reasoning process to the reward signal, effectively increasing the density of gradient information available per rollout. This manifests as significantly higher GSNR for RLTT on these benchmarks, indicating a more informative and coherent learning signal.

On MATH-500, GSNR differences between RLTT and GRPO are small and not statistically significant. This is consistent with the intermediate difficulty of the task: rewards are sufficiently informative for both objectives to produce usable gradients, but not so sparse that credit assignment becomes the dominant optimization bottleneck. In this regime, RLTT neither substantially improves nor degrades gradient consistency relative to GRPO.

GSM8K exhibits a different behavior. RLTT achieves substantially higher accuracy than GRPO on this benchmark, and Table 10 shows a corresponding decrease in GSNR. This reduction does not indicate degraded optimization. Rather, it reflects gradient saturation: once the policy reliably produces correct solutions, gradients become small, reducing the signal-to-noise ratio despite stable training. In this regime, GSNR is no longer a proxy for optimization difficulty, but instead reflects task mastery.

## A.8. Inference Compute Cost Analysis

We additionally analyze the inference-time compute trade-off introduced by looped latent reasoning. Because Ouro performs multiple internal refinement steps before emitting each token, its inference cost is higher than that of a conventional decoder-only transformer with a comparable parameter count. This section quantifies that cost on AIME26 and relates it to the corresponding accuracy gains.

*Table 11.* Inference compute trade-off on AIME26. FLOPs multipliers report the normalized inference FLOPs of Ouro-2.6B-Thinking with four loop iterations relative to the corresponding Qwen3 baseline under the same AIME26 evaluation setting. Accuracies are deterministic Pass@1 results.

| Comparison | Ouro FLOPs Multiplier | Ouro+RLTT AIME26 | Qwen+GRPO AIME26 |
|---|---|---|---|
| Ouro-2.6B vs. Qwen3-1.7B | $6.75\times$ | 26.7% | 0.00% |
| Ouro-2.6B vs. Qwen3-4B | $2.63\times$ | 26.7% | 3.33% |

Table 11 shows that Ouro+RLTT incurs substantially higher per-token FLOPs than the non-looped Qwen3 baselines. Relative to Qwen3-1.7B, Ouro-2.6B requires $6.75\times$ more inference FLOPs per token; relative to Qwen3-4B, it requires $2.63\times$ more inference FLOPs. This increased cost is expected: even though Ouro has fewer parameters than Qwen3-4B, each emitted token is produced after four latent loop iterations, which scales the recurrent transformer computation by approximately the loop count.

However, the added latent compute yields a large improvement on the most difficult AIME-style setting. Ouro+RLTT achieves 26.7% accuracy on AIME26, compared to 0.0% for Qwen3-1.7B+GRPO and 3.33% for Qwen3-4B+GRPO. Thus, the comparison should be interpreted as a compute–accuracy trade-off rather than a pure efficiency comparison: Ouro+RLTT spends more computation per output token, but converts that compute into substantially stronger reasoning performance.

## A.9. Centered Kernel Alignment

A potential failure mode of trajectory-level supervision is overly early convergence: because RLTT directly supervises intermediate loop states, it might force early-loop representations to collapse prematurely toward the final-loop representation, eliminating useful later refinement. We test for this failure mode using a Centered Kernel Alignment (CKA) analysis of latent representations across loop iterations (Kornblith et al., 2019).

For each method, we evaluate the same set of $N = 30$ AIME26 math prompts with exactly four loop iterations. For prompt $e$, let $z_e^{(t)} \in \mathbb{R}^d$ denote the hidden state of the last prompt token after loop $t$. We stack these vectors across prompts to form a dataset-level representation matrix

$$Z^{(t)} = \begin{bmatrix} (z_1^{(t)})^\top \\ \vdots \\ (z_N^{(t)})^\top \end{bmatrix} \in \mathbb{R}^{N \times d}.$$

After centering each $Z^{(t)}$ across prompts, we compute linear CKA between loop representations:

$$\mathrm{CKA}(Z^{(a)}, Z^{(b)}) = \frac{\left\| (\tilde{Z}^{(a)})^\top \tilde{Z}^{(b)} \right\|_F^2}{\left\| (\tilde{Z}^{(a)})^\top \tilde{Z}^{(a)} \right\|_F \left\| (\tilde{Z}^{(b)})^\top \tilde{Z}^{(b)} \right\|_F},$$

where $\tilde{Z}^{(t)}$ denotes the centered representation matrix. We then measure convergence to the terminal loop by computing

$$c_t = \mathrm{CKA}(Z^{(t)}, Z^{(4)}), \qquad t \in \{1, 2, 3\}.$$

*Table 12.* Convergence-to-final CKA analysis across loop iterations. Values report the similarity between each intermediate-loop last-token representation and the final loop-4 representation, computed across 30 math prompts.

| Method | $\mathrm{CKA}(1, 4)$ | $\mathrm{CKA}(2, 4)$ | $\mathrm{CKA}(3, 4)$ | Monotonic? |
|--------|------|------|------|-----------|
| RLTT | 0.849 | 0.869 | 0.978 | Yes |
| GRPO | 0.836 | 0.788 | 0.974 | No |

Table 12 argues against the overly early convergence failure mode. Under RLTT, similarity to the final-loop representation increases monotonically from loop 1 to loop 3. This indicates that intermediate latent states move progressively toward the terminal representation rather than collapsing immediately to it. The increase from loop 2 to loop 3 further suggests that later loop iterations still perform meaningful refinement.

In contrast, GRPO exhibits non-monotonic convergence: similarity to the final-loop representation decreases from loop 1 to loop 2 before recovering at loop 3. This suggests that terminal-only supervision permits intermediate loop states to drift away from the final representational direction before late correction. RLTT therefore appears to stabilize the latent refinement trajectory without eliminating the role of later computation.

Together with the loop-level evaluation in Appendix A.6, these results indicate that trajectory-level supervision improves the usefulness of early latent computation rather than merely collapsing the model toward premature convergence.

## A.10. Theoretical Analysis

**Goal.** We formalize a simple abstraction in which (i) later latent loops correspond to progressively refined next-token distributions, and (ii) RLTT imposes a larger *per-token* uncertainty/variance cost than terminal-only objectives. Under mild diminishing-returns assumptions on how answer correctness improves with additional decoded tokens, we show that the RLTT's induced optimal decoding length is less than or equal to GRPO's.

### A.10.1. SETUP

Fix a prompt $x$. A looped LM generates a response $y = (y_1, \ldots, y_L)$. Let the random variable $L = L(y)$ denotes the response length. For each position $j$ and loop depth $t \in \{1, \ldots, T_{\max}\}$, the model produces a next-token distribution

$$P_\theta^{(t)}(\cdot \mid x, y_{<j}),$$

where $t = T_{\max}$ denotes the terminal-loop distribution.

Let $V(\cdot)$ be any nonnegative *uncertainty / variance functional* on categorical distributions (e.g., entropy or variance). Define the per-token loop uncertainty

$$V_\theta^{(t)}(j) := V\left(P_\theta^{(t)}(\cdot \mid x, y_{<j})\right) \geq 0.$$

Let $\omega_t \geq 0$ be loop weights with $\sum_{t=1}^{T_{\max}} \omega_t = 1$. Define the RLTT trajectory-averaged per-token uncertainty

$$\bar{V}_\theta(j) := \sum_{t=1}^{T_{\max}} \omega_t V_\theta^{(t)}(j).$$

We compare two per-token costs:

$$C_\theta^{\text{GRPO}}(j) := V_\theta^{(T_{\max})}(j), \qquad C_\theta^{\text{RLTT}}(j) := \bar{V}_\theta(j) = \sum_{t=1}^{T_{\max}} \omega_t V_\theta^{(t)}(j).$$

For any fixed policy $\pi_\theta$, define the corresponding total uncertainty costs:

$$L_{\text{total,GRPO}}(\theta) := \mathbb{E}\left[\sum_{j=1}^{L(y)} C_\theta^{\text{GRPO}}(j)\right], \qquad L_{\text{total,RLTT}}(\theta) := \mathbb{E}\left[\sum_{j=1}^{L(y)} C_\theta^{\text{RLTT}}(j)\right],$$

where the expectation is taken over prompts $x$ and rollouts $y \sim \pi_\theta(\cdot \mid x)$.

### A.10.2. ASSUMPTIONS

**Assumption A.1** (Loop refinement decreases uncertainty). For every $j$ and every context $(x, y_{<j})$, uncertainty is non-increasing with loop depth:

$$V_\theta^{(1)}(j) \geq V_\theta^{(2)}(j) \geq \cdots \geq V_\theta^{(T_{\max})}(j).$$

**Assumption A.2** (Diminishing returns of extra decoded tokens). Let $S : \mathbb{Z}_{\geq 0} \to [0, 1]$ denote the best achievable expected terminal reward under a maximum length constraint:

$$S(L) := \sup_{\pi_\theta : \Pr_{y \sim \pi_\theta(\cdot \mid x)}[L(y) \leq L] = 1} \mathbb{E}_{y \sim \pi_\theta(\cdot \mid x)}[r(y)],$$

where $r(y) \in [0, 1]$ depends only on final-answer correctness. We assume that $S(L)$ is non-decreasing and discrete concave in $L$, i.e., the marginal gains

$$\Delta S(L) := S(L + 1) - S(L)$$

are non-increasing in $L$.

**Assumption A.3** (Approximately linear total uncertainty cost). There exist constants $c_{\mathrm{GRPO}}, c_{\mathrm{RLTT}} > 0$ such that, in expectation, each generated response incurs an uncertainty cost approximately linear in response length:

$$\mathbb{E}\left[\sum_{j=1}^{L(y)} C_\theta^{\mathrm{GRPO}}(j) \ \middle| \ L(y) = L\right] \approx c_{\mathrm{GRPO}} L, \qquad \mathbb{E}\left[\sum_{j=1}^{L(y)} C_\theta^{\mathrm{RLTT}}(j) \ \middle| \ L(y) = L\right] \approx c_{\mathrm{RLTT}} L$$

We further assume $c_{\mathrm{RLTT}} \geq c_{\mathrm{GRPO}}$ as $C_\theta^{\mathrm{RLTT}}(j)$ averages per-token uncertainty across all latent loops while $C_\theta^{\mathrm{GRPO}}(j)$ only uses the final-loop uncertainty (thus the induced aggregate uncertainty cost under $C_\theta^{\mathrm{RLTT}}(j)$ is assumed to be no smaller than the induced aggregate uncertainty cost under $C_\theta^{\mathrm{GRPO}}(j)$).

**Remark.** Assumption A.3 abstracts the total uncertainty cost incurred by a generated response as approximately linear in sequence length. This allows us to focus on how differences in aggregate uncertainty cost between RLTT and GRPO affect the optimal decoding length. Our main conclusion relies only on the inequality $c_{\mathrm{RLTT}} \geq c_{\mathrm{GRPO}}$.

A.10.3. STEP 1: RLTT PER-TOKEN COST DOMINATES GRPO PER-TOKEN COST

**Lemma A.4** (Per-token dominance). *Under Assumption A.1, for every token position $j$,*

$$C_\theta^{\mathrm{RLTT}}(j) \ \geq \ C_\theta^{\mathrm{GRPO}}(j).$$

*Proof.* By Assumption A.1, for each $t \in \{1, \ldots, T_{\max}\}$,

$$V_\theta^{(t)}(j) \ \geq \ V_\theta^{(T_{\max})}(j).$$

Multiplying both sides by $\omega_t \geq 0$ and summing over $t$ yields

$$\sum_{t=1}^{T_{\max}} \omega_t V_\theta^{(t)}(j) \ \geq \ \sum_{t=1}^{T_{\max}} \omega_t V_\theta^{(T_{\max})}(j) \ = \ V_\theta^{(T_{\max})}(j),$$

where the last equality uses $\sum_{t=1}^{T_{\max}} \omega_t = 1$. Recognizing the left-hand side as $C_\theta^{\mathrm{RLTT}}(j)$ and the right-hand side as $C_\theta^{\mathrm{GRPO}}(j)$ completes the proof. $\square$

A.10.4. STEP 2: LARGER PER-TOKEN COST IMPLIES SHORTER OPTIMAL LENGTH

We now connect per-token dominance to a shorter *optimal* decoding length by analyzing the abstract reward–cost tradeoff:

$$\max_{L \in \mathbb{Z}_{\geq 0}} \ S(L) \ - \ \lambda c L, \qquad \lambda > 0, \tag{10}$$

where $S(L)$ captures the best achievable expected correctness at length $L$ (Assumption A.2) and $cL$ denotes the expected uncertainty cost of a generated response of length $L$. (Assumption A.3).

Let

$$L_{\mathrm{GRPO}}^* \in \arg\max_{L \geq 0} \ \big(S(L) - \lambda c_{\mathrm{GRPO}} L\big), \qquad L_{\mathrm{RLTT}}^* \in \arg\max_{L \geq 0} \ \big(S(L) - \lambda c_{\mathrm{RLTT}} L\big).$$

**Theorem A.5** (RLTT selects weakly smaller optimal length). *Under Assumptions A.2–A.3 and $c_{\mathrm{RLTT}} \geq c_{\mathrm{GRPO}}$, any optimizer of (10) satisfies*

$$L_{\mathrm{RLTT}}^* \ \leq \ L_{\mathrm{GRPO}}^*.$$

*Proof.* Define discrete marginal gains $\Delta S(L) = S(L+1) - S(L)$. By Assumption A.2, $\Delta S(L)$ is non-increasing in $L$.

Consider extending a decoded sequence of length $L$ by one additional token. Under the objective in (10), this extension is beneficial if the resulting objective value increases, i.e.,

$$S(L+1) - \lambda c(L+1) \ \geq \ S(L) - \lambda cL.$$

Rearranging yields the condition

$$S(L+1) - S(L) \ \geq \ \lambda c,$$

which is equivalently written as $\Delta S(L) \geq \lambda c$.

Because $\Delta S(L)$ is non-increasing in $L$ by Assumption A.2, the optimal decoding length is characterized by a threshold crossing: the length $L$ is increased as long as the marginal reward gain $\Delta S(L)$ exceeds the marginal cost $\lambda c$, and generation stops once $\Delta S(L) < \lambda c$.

Because $c_{\text{RLTT}} \geq c_{\text{GRPO}}$, we have $\lambda c_{\text{RLTT}} \geq \lambda c_{\text{GRPO}}$. Since $\Delta S(L)$ is non-increasing, the first index at which $\Delta S(L)$ drops below the larger threshold $\lambda c_{\text{RLTT}}$ occurs no later than the first index at which it drops below $\lambda c_{\text{GRPO}}$. Therefore any maximizing length under $c_{\text{RLTT}}$ is less than or equal to any maximizing length under $c_{\text{GRPO}}$, i.e. $L^*_{\text{RLTT}} \leq L^*_{\text{GRPO}}$. $\qquad\square$

## A.11. Qualitative Examples

In this section we provide some illuminating examples demonstrating the superior mathematical reasoning induced by RLTT. For clarity, we truncate any additional text generated after the model's first boxed answer.

> ### An Example Where RLTT is Correct Faster
>
> **Question:** A regular octagon is divided into eight regions, labeled $A, B, C, D, E, F, G$, and $H$, by connecting the center point to each vertex. One ball is placed in each region. There are four different colors of balls, and there are infinitely many balls of each color. Balls of the same color may *not* be placed in adjacent regions. How many distinct ways are there to place the balls?
>
> **RLTT:** Okay, so I need to figure out how many ways to place balls in a regular octagon divided into 8 regions, each labeled A to H. The rules are that each region gets one ball, and balls of the same color can't be in adjacent regions. There are four colors, and we have infinitely many of each. Also, rotations and reflections aren't considered the same, so each arrangement is unique even if it's a rotation or reflection of another.
>
> First, let me visualize the octagon. It's divided into 8 regions by connecting the center to each vertex. So each region is a triangle, right? Each region is adjacent to two others. For example, region A is adjacent to B and H, region B is adjacent to A and C, and so on. So the adjacency is like a cycle: A-B-C-D-E-F-G-H-A.
>
> So the problem reduces to coloring a cycle graph with 8 nodes, where each node can be colored with one of four colors, and adjacent nodes can't have the same color. Since rotations and reflections are considered different, we don't have to worry about symmetries reducing the count. So it's just a standard graph coloring problem for a cycle graph $C_8$ with 4 colors.
>
> The formula for the number of colorings of a cycle graph $C_n$ with $k$ colors is
>
> $$(k - 1)^n + (-1)^n (k - 1).$$
>
> Let me recall that. For a cycle with $n$ nodes, the number of proper colorings is
>
> $$(k - 1)^n + (-1)^n (k - 1).$$
>
> So for $n = 8$ and $k = 4$, that would be
>
> $$(4 - 1)^8 + (-1)^8 (4 - 1) = 3^8 + 1 \cdot 3 = 6561 + 3 = 6564.$$
>
> Wait, let me verify that formula. For a cycle graph, the chromatic polynomial is
>
> $$(k - 1)^n + (-1)^n (k - 1).$$
>
> So yes, that's correct. So substituting $n = 8$ and $k = 4$, we get
>
> $$3^8 + 3 = 6561 + 3 = 6564.$$
>
> Therefore, the number of colorings is 6564.
>
> So the answer should be 6564, which in boxed form is $\boxed{6564}$.
>
> **GRPO:** Okay, so I need to figure out how many ways there are to place balls in a regular octagon divided into 8 regions, each labeled A to H. The rules are that each region gets one ball, and balls of the same color can't be in adjacent regions. There are four different colors, and we have infinitely many of each. Also, rotations and reflections aren't considered the same, so each arrangement is unique even if it can be rotated or reflected to look like another.
>
> First, let me visualize the octagon. It's divided into 8 regions by connecting the center to each vertex. So each region is a triangle, right? Each region is adjacent to two others. For example, region A is adjacent to B and H, region B is adjacent to A and C, and so on, up to H, which is adjacent to G and A. So the adjacency is like a cycle: A-B-C-D-E-F-G-H-A.

Since the problem is about coloring regions with four colors such that adjacent regions have different colors, this is similar to a graph coloring problem where the graph is a cycle with 8 nodes. The number of colorings would then be the number of proper colorings of an 8-node cycle graph with 4 colors.

But wait, the problem mentions that rotations and reflections are not considered equivalent. So we don't have to account for symmetries here. That simplifies things because we don't need to use Burnside's lemma or anything like that. Each arrangement is counted as distinct even if it's a rotation or reflection of another.

So, the problem reduces to finding the number of proper colorings of an 8-node cycle graph with 4 colors. The formula for the number of colorings of a cycle graph $C_n$ with $k$ colors is

$$(k-1)^n + (-1)^n (k-1).$$

Let me recall that. For a cycle with $n$ nodes, the chromatic polynomial is

$$(k-1)^n + (-1)^n (k-1).$$

So for $n = 8$ and $k = 4$, that would be

$$(4-1)^8 + (-1)^8 (4-1) = 3^8 + 1 \cdot 3 = 6561 + 3 = 6564.$$

Wait, but let me verify that formula. The chromatic polynomial for a cycle graph $C_n$ is indeed

$$(k-1)^n + (-1)^n (k-1).$$

Let me check for a small $n$. For $n = 3$ (triangle), the chromatic polynomial is $k(k-1)(k-2)$. Plugging into the formula:
$$(k-1)^3 + (-1)^3 (k-1) = (k^3 - 3k^2 + 3k - 1) - (k-1) = k^3 - 3k^2 + 2k,$$

which is $k(k-1)(k-2)$. So that works. For $n = 4$, the chromatic polynomial is

$$(k-1)^4 + (k-1).$$

Let's compute that:
$$(k^4 - 4k^3 + 6k^2 - 4k + 1) + (k-1) = k^4 - 4k^3 + 6k^2 - 3k,$$

which is the same as the chromatic polynomial for a cycle of 4 nodes. So the formula seems correct.
Therefore, applying the formula for $n = 8$ and $k = 4$:

$$(4-1)^8 + (-1)^8 (4-1) = 3^8 + 3 = 6561 + 3 = 6564.$$

So the answer should be 6564.

But wait, let me think again. The problem states that there are four different colors, and we have infinitely many balls of each color. So each region can be colored with any of the four colors, as long as adjacent regions are different. So yes, exactly the cycle graph coloring problem. So the formula applies here.

Alternatively, another way to approach this is using recurrence relations. For a cycle graph, the number of colorings is

$$(k-1)^n + (-1)^n (k-1).$$

For a path graph (which is a line of $n$ nodes), the number of colorings is

$$k (k-1)^{(n-1)}.$$

But since the cycle is a closed loop, the first and last nodes are adjacent, so we have to adjust for that.
Alternatively, using inclusion-exclusion. The total number of colorings without considering the cycle adjacency is

$$k (k-1)^{(n-1)}.$$

But then we have to subtract the cases where the first and last nodes are the same color. Wait, but maybe that's more complicated.

Alternatively, the formula for the number of colorings of a cycle is

$$(k-1)^n + (-1)^n(k-1).$$

So for $n = 8$, $k = 4$, that's

$$3^8 + 3 = 6564.$$

So that's the answer.

But let me check with another approach. Let's model this as a recurrence. Let's denote the number of colorings for a cycle of $n$ nodes as $C(n)$, and for a path (open chain) as $P(n)$. For a path,

$$P(n) = k\,(k-1)^{(n-1)}.$$

For a cycle, $C(n) = P(n) - C(n-1)$. Wait, no, that might not be the case. Wait, actually, the recurrence for cycles is

$$C(n) = (k-1)^n + (-1)^n(k-1).$$

But maybe another way: For a cycle, the first node can be colored in $k$ ways. The second node in $(k-1)$ ways, and so on, but the last node can't be the same as the first or the previous one. So it's similar to the linear case but with an extra constraint.

Alternatively, using the formula for the chromatic polynomial of a cycle:

$$(k-1)^n + (-1)^n(k-1).$$

So for $n = 8$, $k = 4$, that's

$$3^8 + 3 = 6564.$$

So that's the answer.

Alternatively, let's think of it as permutations. For the first region, we have 4 choices. The next region, adjacent to the first, has 3 choices. The third region, adjacent to the second, has 3 choices, and so on. But since it's a cycle, the last region is adjacent to both the first and the seventh region. So the total number would be $4 \cdot 3^7-$ something. Wait, but this approach might not account for the circular dependency.

Wait, actually, the standard formula for the number of colorings of a cycle is indeed

$$(k-1)^n + (-1)^n(k-1).$$

So for $n = 8$, $k = 4$, that's

$$3^8 + 3 = 6564.$$

Therefore, the answer should be 6564.

But let me verify with a smaller $n$. Let's take $n = 3$ (triangle). Then the number of colorings should be

$$4 \cdot 3 \cdot 2 = 24.$$

Using the formula:

$$(4-1)^3 + (-1)^3(4-1) = 27 - 3 = 24.$$

Correct. For $n = 4$, the formula gives

$$3^4 + 3 = 81 + 3 = 84.$$

Let's compute manually: For a cycle of 4 nodes, the number of colorings is $4 \cdot 3 \cdot 2 \cdot 2 + 4 \cdot 3 \cdot 1 \cdot 1$? Wait, no. Wait, for a cycle of 4, the first node has 4 choices, the second 3, the third can't be the same as the second, so 3 choices, but the

fourth can't be the same as the third or the first. If the first and third are the same color, then the fourth has 3 choices. If the first and third are different, the fourth has 2 choices. So total is

$$4 \cdot 3 \left[ 3 \cdot 2 + 1 \cdot 3 \right] = 4 \cdot 3 \left( 6 + 3 \right) = 4 \cdot 3 \cdot 9 = 108.$$

Wait, but according to the formula, it's 84. Hmm, discrepancy here. So maybe my manual calculation is wrong.

Wait, let's do it properly. For a cycle of 4 nodes, the number of colorings is

$$(k-1)^4 + (-1)^4(k-1) = 3^4 + 3 = 81 + 3 = 84.$$

Let's compute it another way. The chromatic polynomial for a cycle of 4 is $k (k-1)^3 - k (k-1)$. Wait, no. Wait, the chromatic polynomial for a cycle is

$$(k-1)^n + (-1)^n(k-1).$$

So for $n = 4$, it's

$$3^4 + 3 = 84.$$

Let's compute it using inclusion-exclusion. The total number of colorings for a path of 4 nodes is

$$4 \cdot 3 \cdot 3 \cdot 3 = 108.$$

But for a cycle, we need to subtract the colorings where the first and last nodes are the same. How many such colorings are there? If the first and last are the same, then it's equivalent to coloring a cycle of 3 nodes (since the first and last are fixed to the same color). Wait, no. Let me think. If we fix the first and last node to the same color, then the second and third nodes must be colored such that they are different from their neighbors. Wait, but the cycle of 3 nodes has

$$4 \cdot 3 \cdot 2 = 24$$

colorings. But if we fix the first and last to the same color, then the number would be

$$4 \text{ (choices for the first/last color)} \cdot 3 \text{ (choices for the second)} \cdot 2 \text{ (choices for the third)} = 24.$$

So the total number of colorings for the cycle would be total colorings for the path (108) minus the colorings where first and last are the same (24), which gives 84. That matches the formula. So the formula works.

Therefore, applying the same logic to $n = 8$, the number of colorings is

$$(4-1)^8 + (-1)^8(4-1) = 3^8 + 3 = 6561 + 3 = 6564.$$

Therefore, the answer should be 6564.

But just to make sure, let's think of another way. Suppose we use recurrence relations. Let's denote $C(n)$ as the number of colorings for a cycle of $n$ nodes, and $P(n)$ as the number for a path. For a path,

$$P(n) = k (k-1)^{(n-1)}.$$

For a cycle, $C(n) = P(n) - C(n-1)$. Wait, no, that might not be correct. Wait, actually, the recurrence for cycles is

$$C(n) = (k-1)^n + (-1)^n(k-1).$$

But maybe another way: For a cycle, the first node has $k$ choices, the second has $(k-1)$, ..., the $n$th node has $(k-2)$ if the first and $n$th are different, but this gets complicated. Alternatively, using the formula for the chromatic polynomial, which is known for cycles.

Alternatively, using the formula from graph theory: The number of proper colorings of a cycle graph $C_n$ with $k$ colors is

$$(k-1)^n + (-1)^n(k-1).$$

So for $n = 8$, $k = 4$, it's

$$3^8 + 3 = 6564.$$

Therefore, the answer is 6564.

I think that's solid. So the final answer is 6564.

$$\boxed{6564}$$

**Another Example Where RLTT is Correct Faster**

**Question:** A steel sphere with a 3-inch radius is made by removing metal from the corners of a cube that has the shortest possible side lengths. How many cubic inches are in the volume of the cube?

**RLTT:** Okay, let's see. The problem is about a steel sphere with a 3-inch radius that's made by removing metal from the corners of a cube. We need to find the volume of the cube with the shortest possible side length. Hmm, so the sphere is inscribed in the cube, right? Because if you remove the corners, the sphere would touch the cube at the centers of each face. Wait, but if the sphere is inscribed, the diameter of the sphere would equal the side length of the cube. Since the radius is 3 inches, the diameter is 6 inches. So the cube's side length is 6 inches. Then the volume would be

$$6^3 = 216$$

cubic inches. But wait, the problem says "the shortest possible side lengths." So maybe there's a different way to inscribe the sphere? No, I think the standard inscribed sphere in a cube has the diameter equal to the cube's edge. So the answer should be 216. Let me double-check. If the sphere is touching the cube's faces, then yes, the cube's edge is twice the radius. So

$$3 \cdot 2 = 6.$$

Volume is

$$6 \cdot 6 \cdot 6 = 216.$$

Yep, that makes sense.

$$\boxed{216}$$

**GRPO:** Okay, let's see. The problem is about a steel sphere with a 3-inch radius that's made by removing metal from the corners of a cube. We need to find the volume of the cube with the shortest possible side length. Hmm, so the sphere is inscribed in the cube, but since they're removing metal from the corners, maybe the sphere touches the cube at certain points?

Wait, if the sphere is made by removing metal from the cube's corners, that probably means the sphere is inscribed in the cube. But if the sphere is inscribed, the diameter of the sphere would equal the side length of the cube. But the sphere has a radius of 3 inches, so the diameter is 6 inches. That would make the cube's side length 6 inches, and the volume would be

$$6^3 = 216$$

cubic inches. But the problem says "the shortest possible side lengths." Is there a way to have a smaller cube?

Wait, maybe I'm misunderstanding. If you remove metal from the corners of a cube to form a sphere, the sphere must fit inside the cube. But if the sphere is touching the cube's faces, then the diameter of the sphere would equal the cube's edge length. But maybe the sphere is not touching the faces but the corners? Wait, no, removing metal from the corners would create a sphere that's tangent to the cube's edges or something else.

Alternatively, perhaps the sphere is inscribed such that it touches the centers of the cube's faces. In that case, the diameter of the sphere would equal the edge length of the cube. So again, 6 inches, volume 216. But the problem says "the shortest possible side lengths." Maybe there's a different configuration where the sphere is not touching the centers but the edges or something else, allowing a smaller cube?

Wait, let's think again. If you have a cube and you remove the corners to make a sphere, the sphere must be tangent to the cube's edges. Wait, but how? Let me visualize. If you take a cube and cut off each corner, the resulting shape is a truncated cube. But the problem says the sphere is made by removing metal from the corners. So maybe the sphere is the largest possible sphere that can fit inside the cube after truncating the corners. But the sphere's radius is given as 3 inches. So we need to find the cube's edge length such that the sphere of radius 3 inches can fit inside the cube after truncating the corners. But how does truncating the corners affect the maximum sphere that can fit?

Alternatively, maybe the sphere is inscribed in the cube, but the cube is the smallest possible such that the sphere can be formed by truncating the cube's corners. Wait, perhaps the sphere touches the midpoints of the cube's edges. Let me

think. If the sphere is tangent to the edges of the cube, then the distance from the center of the cube to the middle of an edge is equal to the sphere's radius.

The cube has edge length $a$. The distance from the center of the cube to the middle of an edge is $(a/2) \cdot \sqrt{2}/2$. Wait, let's calculate that. The center of the cube is at $(a/2, a/2, a/2)$. The middle of an edge would be at $(a/2, 0, a/2)$, for example. The distance from the center to this point is

$$\sqrt{(0)^2 + (a/2)^2 + (0)^2} = a/2.$$

Wait, no. Wait, the middle of an edge is at $(a/2, 0, a/2)$. The center is at $(a/2, a/2, a/2)$. So the distance is

$$\sqrt{(0)^2 + (a/2)^2 + (0)^2} = a/2.$$

So if the sphere is tangent to the edges, then the radius would be $a/2$. But the sphere's radius is 3 inches, so

$$a/2 = 3 \quad \Rightarrow \quad a = 6.$$

Then the volume is

$$6^3 = 216.$$

But that's the same as before. So maybe that's the answer.

But the problem says "the shortest possible side lengths." So if the sphere is tangent to the edges, then the cube's edge length is 6. But maybe there's a way to have a smaller cube where the sphere is tangent to the faces? Wait, if the sphere is tangent to the faces, then the diameter is equal to the edge length, so again $a = 6$. So same result.

Wait, maybe I'm missing something. Let me check again. If the sphere is formed by truncating the cube's corners, perhaps the sphere is not inscribed in the cube but is instead a sphere that fits inside the cube with the corners cut off. But how does truncating the corners affect the maximum sphere that can fit?

Alternatively, maybe the sphere is the largest sphere that can fit inside the cube after truncating the corners. But truncating the corners would make the cube's edges shorter, but the sphere's size would depend on how much you truncate. Wait, but the problem states that the sphere is made by removing metal from the corners of the cube. So the sphere is the result of truncating the cube's corners. So the sphere must fit inside the original cube, and the truncation is such that the sphere is formed. Therefore, the sphere must be inscribed in the cube, touching the centers of the cube's faces. Therefore, the diameter of the sphere equals the edge length of the cube. So edge length is 6, volume 216.

But the problem says "the shortest possible side lengths." So maybe there's a different way to arrange the sphere inside the cube with a smaller edge length. Wait, but if the sphere is 3 inches in radius, then the diameter is 6 inches. So the cube must have at least 6 inches on each side to contain the sphere. If the cube were smaller, the sphere wouldn't fit. Therefore, the minimal cube is $6 \times 6 \times 6$, volume 216. So the answer is 216.

But let me confirm. Suppose the cube has edge length $a$. The sphere has radius 3, so diameter 6. To fit inside the cube, the sphere must fit within the cube. The minimal cube that can contain the sphere is one where the sphere touches the cube's faces. Therefore, the edge length is 6. So volume is

$$6^3 = 216.$$

Therefore, the answer is 216.

But wait, the problem says "made by removing metal from the corners of a cube." So maybe the sphere is not inscribed in the cube, but is formed by truncating the cube's corners. In that case, the sphere would be tangent to the truncated edges. Let me think. If you truncate a cube's corners, you're cutting off each corner, which turns each corner into a new face. The resulting shape is a truncated cube. The sphere that is formed by this truncation would have to fit within the original cube. But how does the truncation affect the sphere's size?

Alternatively, maybe the sphere is the largest sphere that can fit inside the truncated cube. But the problem states that the sphere is made by removing metal from the cube's corners, so the sphere is the result of that truncation. Therefore, the sphere must be tangent to the original cube's edges or something else.

Wait, perhaps the sphere is tangent to the original cube's edges. Let's calculate the distance from the center of the cube to the middle of an edge. As before, that distance is $a/2$. If the sphere is tangent to the edges, then the radius would be $a/2$. So if the radius is 3, then $a = 6$. So again, the cube's edge length is 6, volume 216.

Alternatively, if the sphere is tangent to the original cube's faces, then the radius is $a/2$, so same result. So regardless of whether the sphere is tangent to edges or faces, the edge length is 6. Therefore, the minimal cube is $6 \times 6 \times 6$, volume 216.

Therefore, the answer should be 216 cubic inches.

The steel sphere with a 3-inch radius has a diameter of 6 inches. To fit inside the cube, the cube's side length must equal the sphere's diameter. Thus, the cube has a side length of 6 inches. The volume of the cube is calculated as:

$$6 \times 6 \times 6 = 216.$$

$$\boxed{216}$$

## An Example Where Only GRPO Exhausts Tokens

**Question:** Ten points are randomly and independently generated with a uniform distribution inside a circle. Denote the probability that the center of the circle lies outside the convex hull of these ten points as

$$p = \frac{m}{n},$$

where $m$ and $n$ are coprime. Find

$$n - m$$

**RLTT:** Okay, so I need to find the probability that the center of a circle is outside the convex hull of ten randomly generated points inside the circle. The answer should be in the form $m/n$ where $m$ and $n$ are coprime, and then find $n - m$. Hmm, let's think.

First, the convex hull of points is the smallest convex shape that contains all the points. The center being outside means that all points lie in some semicircle. Wait, no, actually, if all points are contained within a semicircle, then the convex hull would also be within that semicircle, so the center would be outside. So the problem reduces to finding the probability that all ten points lie within some semicircle.

I remember that for $n$ points on a circle, the probability that they all lie within a semicircle is

$$\frac{n}{2^{n-1}}.$$

But wait, is that the case here? Wait, no, that's for points on the circumference. But here the points are inside the circle, not on the circumference. So maybe the problem is different.

Wait, but maybe the same idea applies. If all points are within a semicircle, then the convex hull doesn't contain the center. So the probability that all points lie within some semicircle. For points inside the circle, how do we compute that?

I think the probability that $n$ points lie within a semicircle is

$$\frac{n}{2^{n-1}}.$$

But I need to verify this. Let me think. For points on the circumference, the probability is $\frac{n}{2^{n-1}}$. For points inside the circle, maybe it's the same? Because the radial distance doesn't matter as long as their angles are within a semicircle. So even if they are inside, the angular distribution is uniform. So the probability that all points lie within a semicircle is the same as for points on the circumference. Therefore, the probability $p$ is

$$\frac{10}{2^9} = \frac{10}{512} = \frac{5}{256}.$$

Then $m = 5$, $n = 256$, so
$$n - m = 256 - 5 = 251.$$

Wait, but let me check again. For points inside the circle, the probability that all lie within a semicircle is the same as for points on the circumference? Because the radial component doesn't affect the angular condition. So yes, the probability is $\frac{n}{2^{n-1}}$. So for $n = 10$, it's $\frac{10}{512} = \frac{5}{256}$. Therefore,
$$n - m = 256 - 5 = 251.$$

So the answer is 251.
**Final Answer**

$$\boxed{251}$$

**GRPO:** Okay, so I need to find the probability that the center of a circle is outside the convex hull of ten randomly generated points inside the circle. The answer should be in the form $m/n$ where $m$ and $n$ are coprime, and then find $n - m$. Hmm, let's start by understanding the problem.

First, the convex hull of a set of points is the smallest convex polygon that contains all the points. The center of the circle being outside this convex hull means that all ten points are contained within a semicircle. Wait, is that right? Because if all points are in a semicircle, then the convex hull would also be within that semicircle, so the center wouldn't be inside the convex hull. Conversely, if the convex hull doesn't contain the center, then all points must lie within some semicircle. So the problem reduces to finding the probability that all ten points lie within a semicircle.

But wait, the problem says "the center lies outside the convex hull". So yes, that's equivalent to all points lying within a semicircle. So the probability $p$ is the probability that all ten points lie within some semicircle. Then, the answer is $p = m/n$, and we need to compute $n - m$.

Now, how do we compute this probability? For $n$ points on a circle, the probability that they all lie within some semicircle is
$$\frac{n}{2^{n-1}}.$$

Wait, is that correct? Let me recall. For points on a circle, the probability that all $n$ points lie within a semicircle is
$$\frac{n}{2^{n-1}}.$$

But wait, that's when the points are on the circumference. But in this problem, the points are inside the circle, not on the circumference. So maybe the approach is different.

Wait, but the problem states that the points are generated with a uniform distribution inside the circle. So they can be anywhere inside, not just on the boundary. So the previous formula for points on the circumference might not apply here. Hmm, so I need to adjust my approach.

Let me think. If all points are inside the circle, the convex hull will contain the center if and only if the points are not all contained within a semicircle. So the probability that the center is outside the convex hull is equal to the probability that all ten points lie within some semicircle. So the problem reduces to finding the probability that ten points, uniformly random in the circle, all lie within some semicircle.

But how do we compute that? For points on the circumference, the probability is $n/2^{n-1}$, but for points inside the circle, it's different. Let me check.

I remember that for points inside a circle, the probability that all $n$ points lie within some semicircle is
$$n \left( \frac{1}{2} \right)^{n-1}.$$

Wait, but that seems similar to the circumference case. But maybe not. Wait, actually, when points are inside the circle, the probability might be different because the points can be anywhere, not just on the boundary.

Wait, perhaps the probability is the same as the circumference case? Let me think. If we fix a semicircle, the probability that a single point is inside that semicircle is $1/2$. But since the semicircle can be rotated, the probability that all points

lie within some semicircle is different. For the circumference case, the probability is $n/2^{n-1}$, but for the interior case, maybe it's different.

Alternatively, maybe the probability is the same. Let me check for $n = 2$. For two points inside the circle, the probability that they lie within some semicircle. For two points, the probability that they lie within a semicircle is 1, because you can always rotate the semicircle to contain both points. Wait, but that's not true. If two points are diametrically opposed, then any semicircle containing one point will not contain the other. Wait, but if the points are inside the circle, not on the boundary, then the maximum distance between two points is less than the diameter. So maybe for two points inside the circle, the probability that they lie within some semicircle is 1. Wait, but that can't be. If two points are very close to opposite ends of the circle, then there exists a semicircle that contains both points. Wait, actually, for any two points inside the circle, there exists a semicircle that contains both points. Because you can take the semicircle that is centered at the midpoint between the two points. Wait, but the midpoint might not be the center of the circle. Hmm, maybe not. Let me think.

Suppose two points are placed at $(1, 0)$ and $(-1, 0)$ in a unit circle. Then, any semicircle that is a half-circle from angle $-\pi/2$ to $\pi/2$ would contain both points. Wait, but if the points are at $(1, 0)$ and $(-1, 0)$, then a semicircle from angle 0 to $\pi$ would contain $(1, 0)$ but not $(-1, 0)$. But if we rotate the semicircle, say, from angle $\pi/2$ to $3\pi/2$, then it would contain $(-1, 0)$ but not $(1, 0)$. Wait, but actually, if the semicircle is allowed to be any semicircle, then for two points, there exists a semicircle that contains both. Because you can take the semicircle that is the smallest arc that contains both points. Since the points are inside the circle, the arc between them is less than $\pi$ radians. Wait, but if the two points are diametrically opposed, then the arc between them is $\pi$ radians, so the semicircle would exactly contain them. Wait, but a semicircle is a half-circle, so if the two points are on the boundary, then the semicircle that is the diameter would contain both. But if the points are inside the circle, then even if they are diametrically opposed, you can still have a semicircle that contains both. For example, take a semicircle that is slightly rotated so that it contains both points. Wait, maybe not. Let me visualize.

Suppose two points are at $(1, 0)$ and $(-1, 0)$ in a unit circle. If I take a semicircle from angle $-\pi/2$ to $\pi/2$, that's the right half-circle, which contains $(1, 0)$ but not $(-1, 0)$. If I take a semicircle from $\pi/2$ to $3\pi/2$, that's the left half-circle, which contains $(-1, 0)$ but not $(1, 0)$. But if I take a semicircle that is rotated by $45°$, say from $-\pi/4$ to $3\pi/4$, then does that contain both points? The point $(1, 0)$ is at angle 0, which is within $-\pi/4$ to $3\pi/4$. The point $(-1, 0)$ is at angle $\pi$, which is not within $-\pi/4$ to $3\pi/4$. So that semicircle doesn't contain both. Wait, so maybe for two diametrically opposed points, there is no semicircle that contains both points? But that contradicts my previous thought.

Wait, but if the points are inside the circle, not on the boundary, then they can't be exactly diametrically opposed. Because if they are inside, their distance is less than $2r$. So maybe for two points inside the circle, there always exists a semicircle that contains both. Let me check.

Suppose two points are inside the circle. Let's say the center is $O$. The angle between the two points as viewed from $O$ is $\theta$. If $\theta \leq \pi$, then there exists a semicircle that contains both points. If $\theta > \pi$, then the angle between them is greater than $\pi$, but since they are inside the circle, the maximum angle between two points is $\pi$ (if they are on the boundary). Wait, but if they are inside, the angle can be up to $\pi$.

But in any case, if two points are inside the circle, the angle between them can be up to $\pi$. So if the angle is less than or equal to $\pi$, then there exists a semicircle that contains both points. If the angle is greater than $\pi$, which can't happen because the maximum angle is $\pi$. Wait, no. The angle between two points as viewed from the center is the angle between their position vectors. If the points are inside the circle, the angle can be up to $\pi$. So if the angle is exactly $\pi$, then the two points are diametrically opposed. In that case, can we find a semicircle that contains both? If the semicircle is the one that is the diameter itself, then yes, but a semicircle is a half-circle, so the diameter is a line, not a semicircle. Wait, a semicircle is a $180°$ arc. So if the two points are on the boundary, diametrically opposed, then any semicircle that is a $180°$ arc will contain exactly one of the points. Wait, no. If the semicircle is a $180°$ arc and the two points are diametrically opposed on the boundary, then any semicircle will contain exactly one of the points. Therefore, there is no semicircle that contains both points. Therefore, for two points on the boundary, diametrically opposed, the probability that they lie within some semicircle is zero. But in our problem, the points are inside the circle, not on the boundary. So if the points are strictly inside, then the angle between them can be up to $\pi$, but not exactly $\pi$. Wait, no. If two points are inside the circle, their angle can still be $\pi$ if they are on the boundary. Wait, the boundary is part of the

circle. So if the points are allowed to be on the boundary, then they can be diametrically opposed. But the problem says "inside a circle", which might include the boundary. Hmm, the problem says "inside a circle", so maybe the boundary is included. So in that case, two points can be diametrically opposed on the boundary, and then there is no semicircle that contains both. Therefore, the probability that two points lie within some semicircle is less than 1. But how much less? Wait, this is getting complicated. Maybe I need to look for a general formula for the probability that $n$ points inside a circle all lie within some semicircle. Let me try to recall or derive it.

For points on the circumference, the probability that $n$ points lie within some semicircle is

$$\frac{n}{2^{n-1}}.$$

But for points inside the circle, the probability is different. Let me think.

Suppose we fix a semicircle. The probability that a single point is inside that semicircle is $1/2$. But since the semicircle can be rotated, the probability that all $n$ points lie within some semicircle is the integral over all possible semicircles of the probability that all points are inside that semicircle, divided by the total number of semicircles? Wait, no. Actually, it's a continuous probability, so we need to integrate over all possible angles.

Alternatively, for each point, the probability that it lies within a semicircle is $1/2$, but since the semicircle can be rotated, the events are not independent. So the probability that all $n$ points lie within some semicircle is the probability that there exists an angle $\theta$ such that all points lie within the semicircle from $\theta$ to $\theta + \pi$.

This is similar to the problem of covering the circle with a semicircle. For points on the circumference, the probability is $n/2^{n-1}$, but for points inside the circle, it's different.

Wait, maybe the probability is the same? Let me check for $n = 2$. If the points are inside the circle, the probability that they lie within some semicircle. For two points, the probability that they lie within some semicircle is equal to the probability that the angle between them is less than or equal to $\pi$. Wait, but the angle between them can be up to $\pi$. If the angle is exactly $\pi$, then they are diametrically opposed, and there is no semicircle that contains both. If the angle is less than $\pi$, then there exists a semicircle that contains both. So the probability that two points lie within some semicircle is equal to the probability that the angle between them is less than $\pi$.

But how do we compute that? For two points inside the circle, the angle between them is a random variable

---

**Question:** In the diagram, $D$ and $E$ are the midpoints of $\overline{AB}$ and $\overline{BC}$ respectively. Determine the area of quadrilateral $DBEF$.

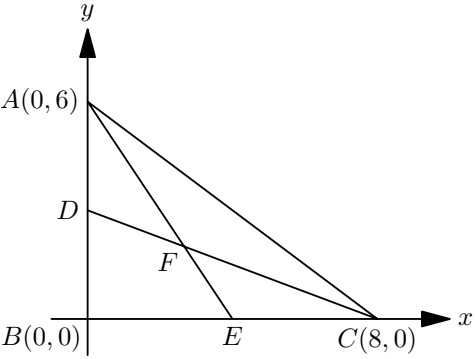

**RLTT:** Okay, let's see. I need to find the area of quadrilateral $DBEF$. The points $D$ and $E$ are midpoints of $\overline{AB}$ and $\overline{BC}$, respectively. The coordinates are given: $A$ is $(0, 6)$, $B$ is $(0, 0)$, $C$ is $(8, 0)$, $D$ is $(0, 3)$, $E$ is $(4, 0)$, and $F$ is $\left(\frac{8}{3}, 2\right)$.

First, maybe I should plot these points to visualize the quadrilateral. $D$ is at $(0, 3)$, $B$ is at $(0, 0)$, $E$ is at $(4, 0)$, and $F$ is at $\left(\frac{8}{3}, 2\right)$. So the quadrilateral is formed by connecting $D$ to $B$ to $E$ to $F$ and back to $D$.

To find the area, one method is to use the shoelace formula. Let me list the coordinates of the quadrilateral in order. The points are $D(0,3)$, $B(0,0)$, $E(4,0)$, $F\left(\frac{8}{3}, 2\right)$. Let's apply the shoelace formula.

The formula is: For points $(x_1, y_1)$, $(x_2, y_2)$, ..., $(x_n, y_n)$, area is

$$\frac{|(x_1 y_2 + x_2 y_3 + \cdots + x_n y_1) - (y_1 x_2 + y_2 x_3 + \cdots + y_n x_1)|}{2}.$$

So plugging in the coordinates:

First part:

$$(0 \cdot 0) + (0 \cdot 0) + (4 \cdot 2) + \left(\frac{8}{3} \cdot 3\right) = 0 + 0 + 8 + 8 = 16.$$

Second part:

$$(3 \cdot 0) + (0 \cdot 4) + \left(0 \cdot \frac{8}{3}\right) + (2 \cdot 0) = 0 + 0 + 0 + 0 = 0.$$

Subtracting:

$$16 - 0 = 16.$$

Take absolute value and divide by 2:

$$\frac{16}{2} = 8.$$

So the area is $8$.

Wait, that seems straightforward. Let me double-check. Maybe I made a mistake in the order of the points. The quadrilateral is $D$-$B$-$E$-$F$-$D$. So the coordinates in order are $(0,3)$, $(0,0)$, $(4,0)$, $\left(\frac{8}{3}, 2\right)$. Yes, that's correct. The shoelace calculation gives 8. So the answer should be 8.

$$\boxed{8}$$

**GRPO:** Okay, let's see. I need to find the area of quadrilateral $DBEF$. The problem says $D$ and $E$ are midpoints of $\overline{AB}$ and $\overline{BC}$ respectively. The Asymptote code gives coordinates for points $A$, $B$, $C$, $D$, $E$, and $F$. Let me first write down all the coordinates to visualize the problem better.

So, $A$ is at $(0,6)$, $B$ is at $(0,0)$, $C$ is at $(8,0)$. $D$ is the midpoint of $\overline{AB}$, so since $A$ is $(0,6)$ and $B$ is $(0,0)$, the midpoint $D$ should be halfway between them. The $x$-coordinate is 0, and the $y$-coordinate is

$$\frac{6+0}{2} = 3.$$

So $D$ is $(0,3)$, which matches the Asymptote code.
$E$ is the midpoint of $\overline{BC}$. $B$ is $(0,0)$ and $C$ is $(8,0)$, so the midpoint $E$ would be at

$$\left(\frac{0+8}{2}, \frac{0+0}{2}\right) = (4,0).$$

That's also given in the code.

$F$ is given as

$$\left(\frac{8}{3}, 2\right).$$

Hmm, I need to figure out where $F$ is. The Asymptote code draws $E - -A - -C - -D$. So, $E$ is connected to $A$, $A$ to $C$, $C$ to $D$, and $D$ to... Wait, the code says `draw(E-A-C-D);` so the lines are $E$ to $A$, $A$ to $C$, $C$ to $D$, and $D$ to... Wait, maybe $D$ is connected back to $E$? But the code doesn't show that. Wait, maybe the figure is $E - A - C - D - E$? But the Asymptote code just draws $E - -A - -C - -D$. So maybe the figure is a polygon $EACD$, but then $D$ is connected back to $E$? Not sure. But the problem mentions quadrilateral $DBEF$. So points $D, B, E, F$. Let me check the coordinates again.

Point $F$ is at
$$\left(\frac{8}{3}, 2\right).$$

Let me see. The Asymptote code draws $E - -A - -C - -D$. So $E$ is $(4, 0)$, $A$ is $(0, 6)$, $C$ is $(8, 0)$, $D$ is $(0, 3)$. So the lines are $E$ to $A$, $A$ to $C$, $C$ to $D$, and $D$ to... Wait, maybe $D$ is connected back to $E$? But the code doesn't show that. Wait, maybe the figure is $E - A - C - D - E$? But the problem says quadrilateral $DBEF$. So points $D, B, E, F$. Let me plot these points mentally.

Point $D$ is $(0, 3)$, $B$ is $(0, 0)$, $E$ is $(4, 0)$, and $F$ is $\left(\frac{8}{3}, 2\right)$. So quadrilateral $DBEF$ is a four-sided figure with vertices at $(0, 3)$, $(0, 0)$, $(4, 0)$, and $\left(\frac{8}{3}, 2\right)$. To find its area, maybe I can use the shoelace formula. Let me recall that the shoelace formula for a polygon with vertices $(x_1, y_1), (x_2, y_2), \ldots, (x_n, y_n)$ is

$$\frac{1}{2}\left|\sum_{i=1}^{n}(x_i y_{i+1} - x_{i+1} y_i)\right|,$$

where $x_{n+1} = x_1$ and $y_{n+1} = y_1$.

So let's list the coordinates of quadrilateral $DBEF$ in order. Let's make sure the order is correct. The quadrilateral is $D - B - E - F - D$. So the coordinates are:

$$D : (0, 3), \quad B : (0, 0), \quad E : (4, 0), \quad F : \left(\frac{8}{3}, 2\right),$$

then back to $D : (0, 3)$.

So applying the shoelace formula:

First, list the coordinates in order:
$$(0, 3), \ (0, 0), \ (4, 0), \ \left(\frac{8}{3}, 2\right), \ (0, 3).$$

Compute the sum of $x_i y_{i+1}$:

First pair: $x_1 = 0, y_1 = 3$; $x_2 = 0, y_2 = 0$,
$$x_1 y_2 = 0 \cdot 0 = 0.$$

Second pair: $x_2 = 0, y_2 = 0$; $x_3 = 4, y_3 = 0$,

$$x_2 y_3 = 0 \cdot 0 = 0.$$

Third pair: $x_3 = 4, y_3 = 0$; $x_4 = \frac{8}{3}, y_4 = 2$,
$$x_3 y_4 = 4 \cdot 2 = 8.$$

Fourth pair: $x_4 = \frac{8}{3}, y_4 = 2$; $x_5 = 0, y_5 = 3$,

$$x_4 y_5 = \frac{8}{3} \cdot 3 = 8.$$

Fifth pair: $x_5 = 0, y_5 = 3$; $x_6 = 0, y_6 = 3$,
$$x_5 y_6 = 0 \cdot 3 = 0.$$

Sum of these terms:
$$0 + 0 + 8 + 8 + 0 = 16.$$

Now compute the sum of $y_i x_{i+1}$:

First pair:
$$y_1 x_2 = 3 \cdot 0 = 0.$$

Second pair:
$$y_2 x_3 = 0 \cdot 4 = 0.$$

Third pair:

$$y_3 x_4 = 0 \cdot \frac{8}{3} = 0.$$

Fourth pair:

$$y_4 x_5 = 2 \cdot 0 = 0.$$

Fifth pair:

$$y_5 x_6 = 3 \cdot 0 = 0.$$

Sum of these terms:

$$0 + 0 + 0 + 0 + 0 = 0.$$

So the area is

$$\frac{1}{2}|16 - 0| = \frac{1}{2} \cdot 16 = 8.$$

Wait, that seems straightforward. But let me verify if the order of the points is correct. Because if the quadrilateral is not convex or if the points are not ordered correctly, the shoelace formula might give the wrong result.

Let me check the order $D - B - E - F - D$.

Starting at $D(0, 3)$, moving to $B(0, 0)$, then to $E(4, 0)$, then up to $F\left(\frac{8}{3}, 2\right)$, which is approximately $(2.666, 2)$, then back to $D(0, 3)$. Hmm, this seems to form a quadrilateral. Let me check if the lines cross each other. From $E(4, 0)$ to $F\left(\frac{8}{3}, 2\right)$, which is moving left and up. Then from $F$ to $D(0, 3)$, which is moving left and up. So the quadrilateral is a four-sided figure without crossing lines. So the order is correct.

Alternatively, maybe I can divide the quadrilateral into triangles or other shapes to compute the area. Let's see. Alternatively, since coordinates are given, maybe using coordinates is the easiest way. The shoelace formula gave me 8. Let me check again.

Wait, let me recalculate the shoelace formula step by step to make sure.
Coordinates in order:

$$D : (0, 3), \quad B : (0, 0), \quad E : (4, 0), \quad F : \left(\frac{8}{3}, 2\right),$$

back to $D : (0, 3)$.

Compute sum of $x_i y_{i+1}$:

$$x_D y_B = 0 \cdot 0 = 0, \quad x_B y_E = 0 \cdot 0 = 0, \quad x_E y_F = 4 \cdot 2 = 8, \quad x_F y_D = \frac{8}{3} \cdot 3 = 8,$$

so

$$0 + 0 + 8 + 8 = 16.$$

Sum of $y_i x_{i+1}$:

$$y_D x_B = 3 \cdot 0 = 0, \quad y_B x_E = 0 \cdot 4 = 0, \quad y_E x_F = 0 \cdot \frac{8}{3} = 0, \quad y_F x_D = 2 \cdot 0 = 0,$$

so

$$0 + 0 + 0 + 0 = 0.$$

Fourth term: $y_4 = 2$, $x_5 = 0$. term4 = $y_4 * x_5 = 2 * 0 = 0$

Fifth term: $y_5 = 3$, $x_6 = 0$. term5 = $y_5 * x_6 = 3 * 0 = 0$

Sum of these terms: $0 + 0 + 0 + 0 + 0 = 0$

So the area is $1/2|16 - 0| = 1/2 * 16 = 8$.

Wait, that seems straightforward. But let me verify if the order of the points is correct. Because if the quadrilateral is not convex or if the points are not ordered correctly, the shoelace formula might give the wrong result.

Let me check the order $D - B - E - F - D$.

Starting at $D(0, 3)$, moving to $B(0, 0)$, then to $E(4, 0)$, then up to $F\left(\frac{8}{3}, 2\right)$, which is approximately $(2.666, 2)$, then back to $D(0, 3)$. Hmm, this seems to form a quadrilateral. Let me check if the lines cross each other. From $E(4, 0)$ to $F\left(\frac{8}{3}, 2\right)$, which is moving left and up. Then from $F$ to $D(0, 3)$, which is moving left and up. So the quadrilateral is a four-sided figure without crossing lines. So the order is correct.

Alternatively, maybe I can divide the quadrilateral into triangles or other shapes to compute the area. Let's see. Alternatively, since coordinates are given, maybe using coordinates is the easiest way. The shoelace formula gave me 8. Let me check again.

Wait, let me recalculate the shoelace formula step by step to make sure.

Coordinates in order:

$D : (0, 3)$
$B : (0, 0)$
$E : (4, 0)$
$F : \left(\frac{8}{3}, 2\right)$

Back to $D : (0, 3)$

Compute sum of $x_i y_{i+1}$:

First term: $x_D \cdot y_B = 0 \cdot 0 = 0$

Second term: $x_B \cdot y_E = 0 \cdot 0 = 0$

Third term: $x_E \cdot y_F = 4 \cdot 2 = 8$

Fourth term: $x_F \cdot y_D = \left(\frac{8}{3}\right) \cdot 3 = 8$

Sum: $0 + 0 + 8 + 8 = 16$

Sum of $y_i x_{i+1}$:

First term: $y_D \cdot x_B = 3 \cdot 0 = 0$

Second term: $y_B \cdot x_E = 0 \cdot 4 = 0$

Third term: $y_E \cdot x_F = 0 \cdot \left(\frac{8}{3}\right) = 0$

Fourth term: $y_F \cdot x$