# OpenReview forum: "Prioritize the Process, Not Just the Outcome: Rewarding Latent Thought Trajectories Improves Reasoning in Looped Language Models"
_ICML.cc/2026/Conference — ICML 2026 regular_

### Official Review · Reviewer_iEjU · 2026-02-27

**Soundness:** 3
**Presentation:** 4
**Significance:** 2
**Originality:** 3
**Overall Recommendation:** 5
**Confidence:** 3

**Summary:**

The authors of the paper proposed RLTT (Reward Latent Thought Trajectories), which backpropagates the credits to every step of latent thoughts and shows the superiority of RLTT over GRPO, which only assigns credit to the last latent thoughts.

The authors trained a 2.6B LLM with RLTT and evaluated the proposed RLTT on four math benchmarks and four non-math benchmarks, and provided a theoretical analysis showing RLTT is more token efficient in reasoning, compared with GRPO.

**Compliance With Llm Reviewing Policy:**

Affirmed.

**Final Justification:**

The rebuttal fully addressed my concerns regarding method properties, theory presentations, and fair comparisons.

**Key Questions For Authors:**

1. *Evaluation setup.* In lines 256 - 265, the author set different token budget for different tasks. I suggest keeping the longest token budgets (e.g., 3072) to prevent criticism of method-specific setups, given that GRPO may use more tokens but still generate the correct result. From the ablation study in Appendix A, I feel 80 is more like the *full potentials* of the GRPO rather than 71.6.

**Limitations:**

yes

**Strengths And Weaknesses:**

**Strength**:
1. The motivation of RLTT sounds and the performance on the benchmarks show the effectiveness of the proposed method.
2. The authors' analysis of why RLTT works and overall presentation is a plus. I can easily follow the logic and perceive the information.

**Weakness**:
1. The change from RLTT and GRPO is straightforward. To make the work more significant, deepdive on what that is specialized in RLTT, such as failure/success mode in different weighting strategies and why AIME24 and GPQA have higher "entropy" in benchmark results in Appendix A.3 further enhances the contribution.
2. I suggest moving more discussions and the definition of the constants c_GRPO and c_RLTT to the main content from the appendix. From my first browsing-style read, the required assumptions are overly strong and weaken Theorem A.5.

---

> ### Author Rebuttal · Authors · 2026-03-31
>
> Thank you for the constructive suggestions. We agree on all three points and can now report further supporting results.
>
> ### Concern 1 — What is specifically distinctive about RLTT beyond the benchmark table?
>
> We ran a Centered Kernel Alignment (CKA)-based analysis of latent representation similarity across loop iterations on AIME26:
>
> | Method | sim(loop 1, loop 4) | sim(loop 2, loop 4) | sim(loop 3, loop 4) | Monotonic? |
> |---|---|---|---|---|
> | **RLTT** | 0.849 | 0.869 | 0.978 | **Yes** |
> | GRPO | 0.836 | 0.788 | 0.974 | No |
>
> Under RLTT, intermediate representations converge monotonically toward the final-loop representation — a signature of stable, progressively refined thinking trajectories. GRPO does not show this property. This clarifies what is specifically induced by RLTT: the model learns to immediately begin forming high-quality latent thoughts rather than relying on late-stage correction. We will also expand the weighting strategy analysis appendix discussion to discuss the higher-variance results on AIME24 and GPQA, attributing these respectively to dataset size (AIME24 has 30 samples, compared to 500/100/1319 samples respectively for MATH500/BeyondAIME/GSM8K) and distribution shift (GPQA is out-of-domain relative to the MATH training set).
>
> ### Concern 2 — Theory presentation (c_GRPO, c_RLTT, Theorem A.5)
>
> We will move the definitions of c_GRPO and c_RLTT, along with the surrounding discussion, into the main paper. We will also reframe Theorem A.5 explicitly as a mechanism-motivated analysis rather than a fully general characterization, consistent with its intended role. We agree the current appendix placement understates the contribution of the theoretical analysis.
>
> ### Concern 3 — Uniform GRPO token budget across all math benchmarks
>
> We agree that a uniform setup is preferable as it eliminates any benchmark-specific confound. All Qwen3+GRPO & Ouro-2.6B+GRPO mathematical reasoning benchmark evaluations are thus done at a 3072-token budget:
>
> | Model | MATH-500 | AIME24 | AIME26 | BeyondAIME | GSM8K |
> |---|---|---|---|---|---|
> | Qwen3-1.7B + GRPO | 53.8% | 0.00% | 0.00% | 0.00% | 80.4% |
> | Qwen3-4B + GRPO | 61.8% | 3.33% | 3.33% | 0.00% | 88.7% |
> | Ouro-2.6B + GRPO | 79.0% | 16.7% | 16.7% | 6.00% |  93.9% |
> | **Ouro-2.6B + RLTT** | **86.0%** | **33.3%** | **26.7%** | **16.0%** | **94.0%** |
>
> We see that RLTT consistently outperforms GRPO on the difficult benchmarks where token budget is a binding constraint and reasoning efficiency matters:  MATH500: 86.0 vs. 79.0, AIME24: 33.3 vs. 16.7, AIME26: 26.7 vs. 16.7, and BeyondAIME: 16.0 vs. 6.0. On GSM8K, RLTT evaluated at 512 tokens and GRPO evaluated at 3072 tokens both reach ceiling performance: 94.0% vs. 93.9%. This near tie still yields an informative comparison as RLTT achieves equivalent performance to GRPO using ~6x fewer tokens - thereby reinforcing the token efficiency argument central to the paper.

---

> > ### Author Rebuttal · Reviewer_iEjU · 2026-03-31
> >
> > Please see my official comment above.

---

> > > ### Author Response · Authors · 2026-04-03
> > >
> > > Thank you for the thoughtful feedback. We truly appreciate the time and effort you invested in reviewing our work!

---

### Official Review · Reviewer_cF1u · 2026-03-09

**Soundness:** 2
**Presentation:** 3
**Significance:** 3
**Originality:** 3
**Overall Recommendation:** 4
**Confidence:** 3

**Summary:**

This paper studies reinforcement learning for Looped Language Models (LoopLMs) and proposes RLTT, a targeted training objective that assigns credit to the entire latent thought trajectory—i.e., the next-token distributions from all internal loops—rather than only the final loop. Empirically, RLTT substantially improves performance over GRPO across multiple reasoning benchmarks and also leads to shorter generations, suggesting improved token efficiency. The authors attribute these gains to two main factors: more token-efficient reasoning at inference time and richer gradient signals during training.

**Compliance With Llm Reviewing Policy:**

Affirmed.

**Final Justification:**

Thanks for the authors’ response. My concerns have been addressed, and I am inclined to maintain my positive score.

**Key Questions For Authors:**

1. Could the authors provide a more detailed analysis of the memory overhead introduced by RLTT?

2. How does RLTT change the intermediate latent representations compared with GRPO?
Since the main claim is that RLTT provides better credit assignment over the latent thought trajectory, I am curious whether the intermediate per-loop representations or next-token distributions become qualitatively or quantitatively different from those learned by GRPO. For example, do the internal states converge earlier, become more stable across loops, or align better with the final correct answer?

**Limitations:**

yes

**Strengths And Weaknesses:**

Strengths：

1. The paper proposes a simple yet effective RL objective for looped transformers. The key idea of assigning reward to the full latent thought trajectory, rather than only the final loop, is intuitive and well aligned with the architecture.

2. The experiments show that RLTT consistently improves performance over the baseline and also leads to shorter generations, suggesting better token efficiency in addition to higher task accuracy.

Weaknesses：
1. The experimental comparison is fairly narrow, as the paper mainly compares RLTT only against GRPO and evaluates it primarily on a single backbone, Ouro-2.6B-Thinking. As a result, the strongest supported conclusion is that RLTT outperforms terminal-only GRPO on this particular class of looped architectures. The paper does not yet establish whether RLTT remains superior to stronger RL baselines or whether it generalizes broadly across other latent-reasoning or looped-transformer models.

2. The paper explicitly notes that the main practical overhead of RLTT is memory rather than compute. Because the method must retain per-loop log-probabilities, the memory cost grows linearly with the number of loops. The authors also report that they had to reduce ppo_max_token_len_per_gpu to 8192, roughly half of the GRPO setting. This raises questions about scalability to longer contexts, larger loop counts, or larger model sizes, which may be especially important for challenging long-horizon reasoning tasks.

---

> ### Author Rebuttal · Authors · 2026-03-31
>
> Thank you for the careful review. We can address your concerns about narrow comparisons, memory overhead, and latent representation differences with concrete results.
>
> ### Concern 1 — Narrow experimental comparison
>
> We have completed Qwen3+GRPO experiments.  As per Reviewer iEjU's suggestion, all Qwen3+GRPO & Ouro-2.6B+GRPO mathematical reasoning benchmark evaluations are done at a 3072-token budget:
>
> | Model | MATH-500 | AIME24 | AIME26 | BeyondAIME | GSM8K | ARC-C | MMLU-ST | GPQA | MBPP |
> |---|---|---|---|---|---|---|---|---|---|
> | Qwen3-1.7B + GRPO | 53.8% | 0.00% | 0.00% | 0.00% | 80.4% | 82.0% | 71.3% | 6.06% | 43.7% |
> | Qwen3-4B + GRPO | 61.8% | 3.33% | 3.33% | 0.00% | 88.7% | 91.5% | 77.3% | 9.09% | 57.6% |
> | Ouro-2.6B + GRPO | 79.0% | 16.7% | 16.7% | 6.00% | 93.9% | 93.7% | 86.1% | 19.7% | 61.3% |
> | **Ouro-2.6B + RLTT** | **86.0%** | **33.3%** | **26.7%** | **16.0%** | **94.0%** | **94.4%** | **89.6%** | **38.4%** | **64.6%** |
>
> Ouro2.6B+RLTT outperforms Qwen3-4B+GRPO across every benchmark despite having fewer parameters — and the advantage is particularly large on the hardest tasks (AIME24: +30.0%, AIME26: +23.4%, BeyondAIME: +16.0%). We acknowledge that the paper's current strongest claim is that RLTT outperforms terminal-only GRPO on Ouro2.6B. The Qwen3+GRPO results now additionally establish that this advantage holds relative to GRPO on larger non-looped models. As touched upon in the reviewer dyMR rebuttal, we restrict evaluation to Ouro because it is the only open-source looped LM with performance competitive with non-looped LM baselines. To probe generalization within the Ouro family, we are also currently running Ouro-1.4B+GRPO & Ouro-1.4B+RLTT experiments, and will report the results in the final manuscript.
>
> ### Concern 2 — Memory overhead
>
> We agree this deserves explicit treatment. These are the key tradeoffs in our current implementation:
>
> | Setting | GRPO | RLTT | Note |
> |---|---|---|---|
> | ppo_max_token_len_per_gpu | 16,384 | 8,192 | RLTT retains per-loop log-probs |
> | Memory scaling with loops | O(1) | O(T_max) | Linear in loop count |
> | Compute overhead vs GRPO | 1.00× | ~1.00× | Weighted sum only |
> | Training time (140 steps) | 54.4 hrs | 49.1 hrs | RLTT 10% faster due to shorter responses |
>
> We will make this tradeoff fully explicit in the final paper. We note that the memory overhead is a property of our current implementation rather than a fundamental constraint of the RLTT objective — checkpointed recomputation over loop iterations is directly applicable and could substantially reduce peak memory overhead. Despite the reduced token packing, RLTT achieves a 10% reduction in total training time relative to GRPO, because its induced response length reduction reduces the total number of tokens processed per step — partially offsetting the memory cost in practice.
>
> ### Concern 3 — How do intermediate latent representations differ under RLTT vs GRPO?
>
> Our Centered Kernel Alignment (CKA) analysis on AIME26 directly addresses this question:
>
> | Method | sim(loop 1, loop 4) | sim(loop 2, loop 4) | sim(loop 3, loop 4) | Monotonic? |
> |---|---|---|---|---|
> | **RLTT** | 0.849 | 0.869 | 0.978 | **Yes** |
> | GRPO | 0.836 | 0.788 | 0.974 | No |
>
> Under RLTT, each token's intermediate loop representation converges monotonically toward its final-loop representation. Under GRPO, similarity actually decreases from loop 2 to loop 3 before recovering — suggesting less stable and less purposeful intermediate computation. This provides direct mechanistic support for the paper's central claim: RLTT induces more effective latent thought trajectories, where intermediate loops representations become progressively more aligned with the final loop representation rather than fluctuating erratically.

---

> > ### Author Rebuttal · Reviewer_cF1u · 2026-04-03
> >
> > Thank you for your detailed response. The rebuttal sufficiently addresses my concerns, and I will maintain my positive score.

---

> > > ### Author Response · Authors · 2026-04-03
> > >
> > > Thank you for the thoughtful feedback. We truly appreciate the time and effort you invested in reviewing our work!

---

### Official Review · Reviewer_dYMR · 2026-03-12

**Soundness:** 3
**Presentation:** 3
**Significance:** 3
**Originality:** 3
**Overall Recommendation:** 4
**Confidence:** 3

**Summary:**

This paper proposes RLTT, a RL framework for looped language models. The authors show that conventional methods like GRPO fail to effectively improve the performance for these models, and instead propose to optimize full latent trajectory by maximizing the probability over chained loop states. They test the performance on both mathematical non-mathematical domains and show significant performance improvement compared to the baselines.

**Compliance With Llm Reviewing Policy:**

Affirmed.

**Final Justification:**

I am maintaining my original positive score.

**Key Questions For Authors:**

1) While the idea is interesting and intuitive, the experimental validation is limited to a single looped model family. Are there other looped or recurrent LM architectures or families on which RLTT can be evaluated?

2)  It would also be helpful if the authors could include the performance for GRPO on conventional autoregressive RL baselines (i.e. Qwen or DeepSeek) to better understand whether this direction is competitive not only in length reduction but also competitive with conventional LM + RL settings. While apple-to-apple performance would be difficult, I think such analysis could help clarify the broader motivation for pursing looped LMs and specialized RL on that setting.

3) Looped transformers introduce additional per-token computation because of repeated latent refinements. I am curious as to how the overall resulting inference computation cost would differ with the baselines (including autoregressive LMs).

4) While the theory in appendix nicely explains why RLTT may lead to shorter length and early exit, but seems to not fully resolve the following question: Could trajectory-level supervision create a sort of tension with the purpose of looped models with later refinements by encouraging overly early convergence? (i.e. the objective here is from the first loop to output the terminal distribution)

**Limitations:**

yes

**Strengths And Weaknesses:**

Strengths
1) The proposed method is simple and intuitive.
2) The proposed method shows strong empirical performance on both mathematical and non-mathematical benchmarks.
3) The authors provide useful analysis of generation behavior such as reduced response length and lower output entropy with some theoretical explanation.

Weaknesses
1) Although the comparisons against standard LM baselines vs. Ouro-based baselines are useful, it is unclear as to what benefit it can give compared to the LM baselines + GRPO. The paper may be stronger by including a controlled comparison against non-looped LMs post-trained with GRPO.

2) The authors test RLTT with a single lopped backbone, Ouro-2.6B-Thinking, thereby making difficult to determine whether the method can generalize across other looped language models.

---

> ### Author Rebuttal · Authors · 2026-03-31
>
> Thank you for the thoughtful feedback. We can now provide concrete results on all four concerns you raised.
>
> ### Concern 1 — Generalization beyond a single looped backbone
>
> We restrict evaluation to Ouro because it is the only open-source looped LM with performance competitive with non-looped LM baselines. To probe generalization within the Ouro family, we are currently running GPRO/RLTT experiments on the smaller Ouro-1.4B architecture, and will report results in the revised manuscript.
>
> ### Concern 2 — Comparison against non-looped LMs trained with GRPO
>
> We have completed Qwen3+GRPO experiments.  As per Reviewer iEjU's suggestion, all Qwen3+GRPO & Ouro-2.6B+GRPO mathematical reasoning benchmark evaluations are done at a 3072-token budget:
>
>
> | Model | MATH-500 | AIME24 | AIME26 | BeyondAIME | GSM8K | ARC-C | MMLU-ST | GPQA | MBPP |
> |---|---|---|---|---|---|---|---|---|---|
> | Qwen3-1.7B + GRPO | 53.8% | 0.00% | 0.00% | 0.00% | 80.4% | 82.0% | 71.3% | 6.06% | 43.7% |
> | Qwen3-4B + GRPO | 61.8% | 3.33% | 3.33% | 0.00% | 88.7% | 91.5% | 77.3% | 9.09% | 57.6% |
> | Ouro-2.6B + GRPO | 79.0% | 16.7% | 16.7% | 6.00% | 93.9% | 93.7% | 86.1% | 19.7% | 61.3% |
> | **Ouro-2.6B + RLTT** | **86.0%** | **33.3%** | **26.7%** | **16.0%** | **94.0%** | **94.4%** | **89.6%** | **38.4%** | **64.6%** |
>
> Ouro-2.6B+RLTT outperforms Qwen3-4B+GRPO on every benchmark despite having fewer parameters. The gap is most pronounced on hard reasoning tasks — AIME24 (+30.0%), AIME26 (+23.4%), BeyondAIME (+16.0%), and GPQA (+29.3%). These results suggest that RLTT+looped LMs could serve as a promising alternative to GRPO+standard LMs.
>
> ### Concern 3 — Inference compute cost
>
> We have conducted an inference FLOPs analysis over AIME26:
>
> | Comparison | Ouro FLOPs multiplier | Ouro+RLTT AIME26 | Qwen+GRPO AIME26 |
> |---|---|---|---|
> | Ouro-2.6B vs Qwen3-1.7B | 6.75× | 26.7% | 0.00% |
> | Ouro-2.6B vs Qwen3-4B | 2.63× | 26.7% | 3.33% |
>
> Ouro incurs 2.63× the FLOPs of Qwen3-4B at inference (a direct consequence of performing 4 loop iterations per forward pass, which scales inference FLOPs by ~4× the base transformer cost) but achieves substantially higher AIME26 accuracy (26.7% vs 3.33%). We will add this analysis to the paper and explicitly frame RLTT's gains as trading additional per-token compute for substantially improved reasoning quality and shorter total generation length.
>
> ### Concern 4 — Does trajectory-level supervision encourage overly early convergence?
>
> Our Centered Kernel Alignment (CKA)-based analysis of latent representations across loop iterations argues against this failure mode:
>
> | Method | sim(loop 1, loop 4) | sim(loop 2, loop 4) | sim(loop 3, loop 4) | Monotonic? |
> |---|---|---|---|---|
> | **RLTT** | 0.849 | 0.869 | 0.978 | **Yes** |
> | GRPO | 0.836 | 0.788 | 0.974 | No |
>
> Under RLTT, intermediate loop representations converge toward the final-loop representation in a consistent, monotonically increasing manner. Under GRPO, similarity actually decreases from loop 2 to loop 3 before recovering — suggesting less stable intermediate computation. RLTT also outperforms GRPO at every loop budget evaluated (1 to 4), indicating that trajectory-level supervision improves the usefulness of early latent computation rather than collapsing later refinement.

---

> > ### Author Rebuttal · Reviewer_dYMR · 2026-04-01
> >
> > I'd like to thank authors for the detailed response. I will maintain my positive score.

---

> > > ### Author Response · Authors · 2026-04-03
> > >
> > > Thank you for the thoughtful feedback. We truly appreciate the time and effort you invested in reviewing our work!

---

### Official Review · Reviewer_kpUH · 2026-03-13

**Soundness:** 3
**Presentation:** 3
**Significance:** 3
**Originality:** 3
**Overall Recommendation:** 5
**Confidence:** 3

**Summary:**

This paper introduces RLTT (Reward Latent Thought Trajectories) to address training challenges in Looped Language Models (LoopLMs) when using reinforcement learning. By distributing the learned reward signals across every iteration rather than solely at the final loop, RLTT enhances the effectiveness of RL training. Experimental results have demonstrated the efficacy of this method, and the paper further provides theoretical analysis alongside extensive analytical experiments.

**Compliance With Llm Reviewing Policy:**

Affirmed.

**Final Justification:**

I thank the authors for their detailed response, which has addressed my concerns, and I will maintain my positive score.

**Key Questions For Authors:**

See Weaknesses.

**Limitations:**

yes

**Strengths And Weaknesses:**

Strengths:
1. Proposes a new algorithm to improve the effectiveness of reinforcement learning training for Looped Language Models.
2. Conducts extensive experiments along with thorough analytical studies.
3. Provides theoretical derivations explaining that the method boost the token-efficient reasoning in Looped Language Models.

Weaknesses:
1. More mathematical tasks could be evaluated, as the current testing datasets are relatively limited.
2. The paper does not compare with more training methods, such as PPO or variants of GRPO.

---

> ### Author Rebuttal · Authors · 2026-03-31
>
> Thank you for these helpful suggestions. To address both concerns directly, we have added a broader math evaluation and an additional RL baseline comparison.
>
> ### Concern 1 — Broader mathematical evaluation
>
> We have added AIME26 to our evaluation. The results are consistent with the rest of the paper and show the same pattern of Ouro+RLTT superiority over other methods:
>
>
> | Model | AIME26 |
> |---|---|
> | Qwen3-1.7B | 0.00% |
> | Qwen3-4B | 3.33% |
> | DeepSeekR1-1.5B | 3.33% |
> | DeepSeekR1-7B | 6.67% |
> | Ouro-2.6B | 6.67% |
> | Ouro-2.6B+SFT | 13.3% |
> | Ouro-2.6B+GRPO | 16.7% |
> | **Ouro-2.6B+RLTT** | **26.7%** |
>
> Ouro-2.6B+RLTT achieves 26.7% on AIME26, outperforming Ouro-2.6B+GRPO (16.7%), Ouro-2.6B+SFT(13.3%), and the base Ouro-2.6B (6.67%). Ouro-2.6B also outperforms the larger DeepSeekR1-7B and Qwen3-4B non-looped LM baselines. We are currently running repeated evaluations and significance testing to firm up these findings and will report them in the revised manuscript.
>
> ### Concern 2 — Additional RL baselines
>
> We have completed Qwen3+GRPO experiments, which allow direct comparison of RLTT against GRPO applied to a non-looped LM. As per Reviewer iEjU's suggestion, all Qwen3+GRPO & Ouro-2.6B+GRPO evaluations are done at a 3072-token budget:
>
> | Model | MATH-500 | AIME24 | AIME26 | BeyondAIME | GSM8K |
> |---|---|---|---|---|---|
> | Qwen3-1.7B + GRPO | 53.8% | 0.00% | 0.00% | 0.00% | 80.4% |
> | Qwen3-4B + GRPO | 61.8% | 3.33% | 3.33% | 0.00% | 88.7% |
> | Ouro-2.6B + GRPO | 79.0% | 16.7% | 16.7% | 6.00% | 93.9% |
> | **Ouro-2.6B + RLTT** | **86.0%** | **33.3%** | **26.7%** | **16.0%** | **94.0%** |
>
> Ouro-2.6B+RLTT outperforms Qwen3-4B+GRPO across every math benchmark despite operating at fewer parameters. On the hardest benchmarks — AIME24, AIME26, and BeyondAIME — Qwen3-4B+GRPO achieves 3.33%, 3.33%, and 0.00% respectively, compared to 33.3%, 26.7%, and 16.0% for Ouro-2.6B+RLTT. These results suggest that RLTT applied to Ouro-2.6B is not only superior to GRPO applied to Ouro-2.6B, but also competitive with GRPO applied to a larger non-looped LM. To probe generalization within the Ouro family, we are also currently running Ouro-1.4B+GRPO & Ouro-1.4B+RLTT experiments, and will report the results in the final manuscript.
>
> It should be noted that we focus on Ouro+GRPO as the primary baseline because it most directly isolates the paper’s central question: whether trajectory-level credit assignment improves RLVR for LoopLMs relative to a terminal-loop objective on the same architecture. We view PPO and GRPO variants as follow-up extensions rather than necessary baselines for establishing this core comparison, and will clarify this scope in the revision.

---

> > ### Author Rebuttal · Reviewer_kpUH · 2026-04-03
> >
> > I thank the authors for their detailed response, which has addressed my concerns, and I will maintain my positive score.

---

> > > ### Author Response · Authors · 2026-04-03
> > >
> > > Thank you for the thoughtful feedback. We truly appreciate the time and effort you invested in reviewing our work!

---

### Decision · Program_Chairs · 2026-04-30

**Decision:**

Accept (regular)

**Comment:**

This submission introduces RLTT: a RL objective for looped language models that assigns credits across the full latent thought trajectory than assignments only at the final loop. The reviewers agree that RLTT is simple and architecture-aligned, showing convincing empirical improvement over GRPO, while also showing better token efficiency.

The indicated limitation is the breadth of validation than the core technical idea. Reviewers agree that the experiments centered on a single looped backbone, making it difficult to assess generalization across other looped or latent reasoning architectures. The baseline comparisons are also somewhat narrow, since the paper mainly shows improvement over GRPO in the looped setting rather than against a broader set of RL methods or conventional autoregressive LMs trained with RL. Additional concerns include memory overhead and limited discussions.

However the submission identifies a genuine mismatch between standard RL credit assignment and the internal iterative structure of looped models.Then it addresses with an intuitive and effective objective. Reviewers agree that the method technically sound, the empirical gains meaningful across both math and non-math tasks, and the analysis of shorter, lower-entropy, more token-efficient reasoning particularly valuable. Also, the rebuttal appears to have addressed most reviewer concerns, and all reviewers remained positive in their final assessments.

All in all, I recommend accept suggesting updates on the draft based on the additional clarifications provided on author-reviewer discussions.